# Seeking Global Flat Minima in Federated Domain Generalization via Constrained Adversarial Augmentation

## Abstract

Federated domain generalization (FedDG) aims at equipping the federally trained model with the domain generalization ability when the model meets new clients with domain shifts. Among factors that possibly indicate generalization, the loss landscape flatness of the trained model is an intuitive, viable, and widely studied one. However, pursuing the flatness of the global model in the FedDG setting is not trivial due to the restriction to preserve data privacy. To address this issue, we propose GFM, a novel algorithm designed to seek Global Flat Minima of the global model. Specifically, GFM leverages a global model-constrained adversarial data augmentation strategy, creating a surrogate for global data within each local client, which allows for split sharpness-aware minimization to approach global flat minima. GFM is compatible with federated learning without compromising data privacy restrictions, and theoretical analysis further supports its rationality by demonstrating that the objective of GFM serves as an upper bound on the robust risk of the global model on global data distribution. Extensive experiments on multiple FedDG benchmarks demonstrate that GFM consistently outperforms previous FedDG and federated learning approaches.

## 1 Introduction

In recent years, federated learning has emerged as a popular paradigm for distributed learning with data privacy preservation (Kairouz et al., 2021; Li et al., 2020; McMahan et al., 2017). In federated learning, distributed clients keep their data locally and no data are shared across clients. The clients collaborate on training the global model with the intervention of a central server. In each communication round, clients train their local models on their respective datasets and upload them to the server. Then, the server aggregates these models to derive a global model, which is subsequently distributed to all clients. In this way, the global model performs well on clients participating in the training. However, in real scenarios, the federally-trained model may be deployed for clients which don't participate in the training and may experience domain shifts. This challenges the generalization ability of the trained model, which is known as the federated domain generalization problem.

The challenge of federated domain generalization has garnered significant attention in recent year (Guo et al., 2023b; Zhang et al., 2023a;b; Nguyen et al., 2022; Park et al., 2024). Most promising methods try to align the behaviors of local models from various perspectives. To give a few examples, Zhang et al. (2023a) proposed aligning the feature distribution, Guo et al. (2023b) aimed to learn domain-invariant representations by aligning the gradients, and Park et al. (2024) enabled style sharing among different clients. In contrast to these studies, we concentrate more on the optimization solution of the global model from the perspective of loss landscape flatness. There is substantial body of literature (Chen et al., 2021; Izmailov et al., 2018; Jastrzębski et al., 2018; Keskar et al., 2016) on the relationship between loss landscape flatness and the model's generalization ability. Moreover, empirical results in many centralized tasks illustrate the effectiveness of seeking flat minima, including i.i.d. situations (Keskar et al., 2016; Izmailov et al., 2018; Foret et al., 2020), centralized domain generalization (Cha et al., 2021), and incremental learning (Shi et al., 2021a).

However, in federated learning, the flatness of the global model is difficult to estimate and optimize due to privacy concerns, making it a challenging problem. Some studies like FedSAM (Caldarola

et al., 2022; Qu et al., 2022) bypassed this issue by instead focusing on seeking flat minima of local models, hoping this would facilitate the flatness of the global model. However, it inevitably results in sub-optimal solutions. These methods achieve their objective by employing the Sharpness-Aware Minimizer (SAM (Foret et al., 2020)) during local updates on client models. To address the limitations of local flatness methods, FedGAMMA (Dai et al., 2023) introduced global information into local updates by correcting local gradients, ensuring that all clients adjust their updates toward the global direction. However, this gradient correction is not explicitly connected to flatness, and the SAM optimizer is still applied locally without modifications, making it fundamentally a method for local flatness. FedSMOO (Sun et al., 2023) turned to enforce high consistency in local SAM perturbations by approximating the global perturbation using ADMM. However, the approximation is not strict, as the global perturbation is only computed in each round but required in every iteration. Alternatively, Li et al. (2023) explored aggregation weights and demonstrated that weight shrinking leads to flatter global minima. Nevertheless, their method relies on an additional proxy dataset to determine the parameters, which may not always be feasible.

Given that the challenge arises from the lack of direct access to global data, we try to solve it in a data-centric manner by decomposing the objective of seeking global flatness into two components: seeking local flatness and enhancing global-local consistency. We begin with the homogeneous setting, where the data from each local client lies in the same global data distribution. We show that if local models are averaged in a convex combination, the robust risk of the global model is upper-bounded by the convex combination of robust risks of the local models. This suggests that seeking global flat minima by asking for local flatness is practically reasonable if clients are homogeneous. However, the homogeneous assumption does not hold in FedDG and the only source of global information is the global model itself. Therefore, we propose a global model-constrained adversarial data augmentation strategy to augment local data. The augmented data serves as a surrogate for global data, thereby enhancing global-local consistency. These two schemes collaborate on the same goal of approaching global flat minima, each playing a different role: the local flatness objective contributes to the "flatness" of the global model, while the global model-constrained adversarial data augmentation strategy supplements information of the "global" data distribution. Furthermore, theoretical analysis provides additional support for the validity of the proposed method by demonstrating that the objective of GFM provides an upper bound to the robust risk of the global model on the global data distribution. The main contributions of our work are summarized as follows:

- We propose a novel algorithm, GFM, which is specifically designed to seek global flat minima in the federated learning task, which improve domain generalization performance while simultaneously maintaining data privacy.
- We have theoretically demonstrated that the objective of GFM constitutes a component of the upper bound of the risk in the unseen domain. This is evidenced by indicating that the robust empirical risks of local clients on augmented samples is an upper bound of the robust risk of the global model on global data distribution.
- Through extensive experiments on a range of benchmarks, we show that our algorithm can achieve consistently improved performance compared to previous SOTA methods.

## 2 PRELIMINARIES

### 2.1 PROBLEM FORMULATION

The federated domain generalization task aims to train a model that exhibits generalization performance across both seen and unseen domains, adhering to the principles of privacy-preservation inherent in federated learning. A domain is deemed "seen" if a client belonging to it participates in the federated training procedure and vice versa. We denote the set of seen domains during training as $\mathcal{D}^s = \{D_i^s\}_{i=1}^{M_s}$, the set of unseen domains as $\mathcal{D}^u = \{D_i^u\}_{i=1}^{M_u}$, and the set of all domains as $\mathcal{D} = \mathcal{D}^s \cup \mathcal{D}^u$. The data of client $i$ comes from the domain $D_i$ ($D_i \in \mathcal{D}$) and the sampling of data follows: $(x, y) \sim D_i \subset \mathcal{X} \times \mathcal{Y}$. The model to be trained is referred to as $f(\cdot; \theta) : \mathcal{X} \to \mathcal{Y}$, which takes $x$ as input and outputs the prediction for $y$, parameterized by $\theta$. Formally, given a loss function $\ell : \mathcal{Y} \times \mathcal{Y} \to \mathbb{R}$ measuring the discrepancy of the prediction and the label, the ideal objective is as follows:

$$\min_{\theta} \mathcal{E}_{\mathcal{D}}(\theta) := \frac{1}{M_s + M_u} \sum_{D \in \mathcal{D}} \mathbb{E}_{(x,y) \sim D} \ell(f(x; \theta), y). \tag{1}$$

In the FedDG setting, only seen clients are involved in training. Thus, the empirical objective is:

$$\min_{\theta} \hat{\mathcal{E}}_{\mathcal{D}_s}(\theta) := \sum_{D_i^s \in \mathcal{D}^s} p_i \sum_{(x,y) \in \hat{D}_i^s} \frac{1}{|\hat{D}_i^s|} \ell(f(x;\theta), y), \tag{2}$$

where $\hat{D}_i^s$ is the dataset of the $i$-th seen client sampled from $D_i^s$ and $p_i = |\hat{D}_i^s|/\sum_j |\hat{D}_j^s|$. The gap between the practical and the ideal objective reveals the first difficulty of FedDG, wherein the model is required to generalize to the unseen domains by learning knowledge from only data in seen domains. The second challenge lies in the difficulty for the model to explicitly learn the invariant relationship across different domains due to data privacy concerns. That is, each client preserves its own data, which results in no data from different domains being observed simultaneously at a single client. This inevitably leads to an overfitting trend to the local data domain during the local training stage. How to aggregate information from different seen local data distributions with federated principles and ensuring the model's generalization ability to unseen domains remains a challenge.

## 2.2 RELATIONSHIP BETWEEN FLATNESS AND DOMAIN GENERALIZATION

The practical objective in Eq. (2) may have multiple solutions with similar values but different flatness. Intuitively, the model with a flat minimum is more robust to distribution shifts and exhibits better generalization capabilities. However, the commonly used optimizers in the training of deep models tend to find sharp and shallow optima (Keskar et al., 2016), which is significant under the federated situation (Caldarola et al., 2022). In the context of domain generalization, the impact appears to be more severe due to the large domain shift. In this paper, we aim to seek flat minima by minimizing the robust empirical risk, defined as:

$$\hat{\mathcal{E}}^{\gamma}(\theta) := \max_{||\Delta|| < \gamma} \hat{\mathcal{E}}(\theta + \Delta), \tag{3}$$

where $\gamma$ denotes the radius defining a neighborhood around $\theta$. A larger robust risk indicates the presence of a direction within the neighborhood along which the empirical risk increases. The robust risk directly relates to both flatness and optimality of $\theta$ when $\theta$ is a local minimum. To theoretically understand the relationship between flatness and domain generalization, Cha et al. (2021) proposed Theorem 1, which assumes a single test (unseen) domain $T$ and the equal number of samples in each domain. One can see Appendix E for the proof and other details.

**Theorem 1.** *Consider a set of $K$ covers $\{\Theta_k\}_{k=1}^K$ such that the parameter space $\Theta \subset \cup_k^K \Theta_k$ where $diam(\Theta) := \sup_{\theta, \theta' \in \Theta} \|\theta - \theta'\|_2$, $K := \left\lceil (diam(\Theta)/\gamma)^d \right\rceil$ and $d$ is dimension of $\Theta$. Let $v_k$ be a VC dimension of each $\Theta_k$. Then, for any $\theta \in \Theta$, the following bound holds with probability at least $1 - \delta$,*

$$\mathcal{E}_T(\theta) \leq \hat{\mathcal{E}}_{\mathcal{D}^s}^{\gamma}(\theta) + \frac{1}{2M_s} \sum_{i=1}^{M_s} \mathbf{Div}(D_i, T) + \max_{k \in [1,K]} \sqrt{\frac{v_k \ln(n/v_k) + \ln(K/\delta)}{n}}, \tag{4}$$

*where $\mathcal{D}^s$ is the set of train (seen) domains, $n$ is the number of training samples per domain, and $\mathbf{Div}(D_i, T) := 2\sup_A |\mathbb{P}_{D_i}(A) - \mathbb{P}_T(A)|$ is a divergence between two distributions.*

Theorem 1 indicates that the risk $\mathcal{E}_T(\theta)$ on the unseen domain $T$ is upper bounded by the robust empirical risk $\hat{\mathcal{E}}_{\mathcal{D}}^{\gamma}(\theta)$ on the mixture of seen domain $D$, the sum of discrepancy between each seen domain and the test domain, and confidence bound. As a result, the performance on the unseen domains is directly related to the flatness of the seen domains.

## 3 METHOD

Taking Theorem 1 into consideration, we hypothesize seeking a flat optimal solution can ameliorate the generalization performance, which is not satisfied in heterogeneous federated learning tasks according to (Caldarola et al., 2022) and our experiments in Sec. 4.3. However, it is not trivial to directly train a flat global model due to the data privacy concern. As a result, we propose GFM to split the minimization of $\hat{\mathcal{E}}_{\mathcal{D}}^{\gamma}(\theta)$ to local clients. To achieve this goal, we first split the seeking of flatness in the global model into local models by assuming aggregation helps generalization.

Then, to avoid the requirement of global data distribution, we propose a global model-constrained adversarial data augmentation strategy. By combining these two parts, one can directly minimize the upper bound of $\hat{\mathcal{E}}_{\mathcal{D}}^{\gamma}(\theta)$ in local clients to seek global flatness.

## 3.1 SPLIT THE SEEKING OF FLATNESS TO LOCAL MODELS

The objective of seeking global flatness is to minimize the empirical robust risk $\hat{\mathcal{E}}_{\mathcal{D}}^{\gamma}(\theta)$ as follows:

$$\min_{\theta} \hat{\mathcal{E}}_{\mathcal{D}}^{\gamma}(\theta) = \min_{\theta} \max_{||\Delta|| < \gamma} \sum_{D_i^s \in \mathcal{D}^s} p_i \frac{1}{|\hat{D}_i^s|} \sum_{(x,y) \in \hat{D}_i^s} \ell(f(x; \theta + \Delta), y), \tag{5}$$

where $\theta$ refers to the global model. This objective can't be directly calculated in the federated learning setting for two main reasons. First, the inner maximization step needs the gradients of the global model which is hard to estimate during local updates. Second, the gradients are supposed to be calculated on the global data which is not available for local clients. To that effect, we relax the objective in GFM by its upper bound with the following assumption.

**Assumption 1.** *If (1) data distributions $\{D_i\}_{i=1}^{M_S}$ across clients exhibit a non-trivial degree of heterogeneity, and (2) each client has access to a sufficiently large local dataset to estimate the data distribution. Then during the training phase, local models $\{\theta_i\}_{i=1}^{M_S}$ and their aggregate $\sum_i p_i \theta_i$, when weighted by coefficients specific to clients, satisfy the following inequality:*

$$\hat{\mathcal{E}}_D(\sum_i p_i \theta_i) \leq \sum_i p_i \hat{\mathcal{E}}_D(\theta_i), \tag{6}$$

*where $p_i$ represents the coefficient of client $i$.*

Assumption 1 focuses on the change in global risk before and after model aggregation, based on the intuition that aggregating models enhances generalization, which aligns with common practices. Furthermore, if Assumption 1 does not hold, it would imply that at least one local model outperforms the global model (i.e., with lower risk). This suggests that training on a specific domain could result in performance improvements across all domains, which appears counterintuitive in the context of FedDG, where each client's data is restricted to a single domain. It is worth noting that Eq. (6) shares a similar structure with the convex basin assumption proposed in linear connectivity studies (Entezari et al., 2021; Juneja et al., 2022). The convex basin assumption is stricter, as it considers all convex combination coefficients $\{p_i\}_{i=1}^{M_s}$, while in federated learning, $p_i$ is usually fixed and relevant to the number of training samples. In contrast, Assumption 1 is a mild assumption that empirically holds during the federated training process (see more details in Sec. 4.4). With Assumption 1, we derive the following upper bound:

$$\hat{\mathcal{E}}_D^{\gamma}(\theta) \leq \sum_i p_i \hat{\mathcal{E}}_D^{\gamma}(\theta_i) = \sum_i p_i \max_{||\Delta_i|| < \gamma} \frac{1}{|\hat{D}|} \sum_{(x,y) \in \hat{D}} \ell(f(x; \theta_i + \Delta_i), y), \tag{7}$$

where $\hat{D} = \bigcup_i \hat{D}_i^s$. Thus, the global objective is split into multiple local objectives as follows:

$$\min_{\theta_i} \max_{||\Delta_i|| < \gamma} \frac{1}{|\hat{D}|} \sum_{(x,y) \in \hat{D}} \ell(f(x; \theta_i + \Delta_i), y). \tag{8}$$

Eq. (8) indicates that the flatness of the local models on the global data distribution serves as an upper bound for the flatness of the global model, providing a method to seek global flatness through local updates. However, since the global data distribution is not accessible, we resort to seeking a surrogate. Notably, regularization-based methods can be applied in the absence of global data, though they are sub-optimal since they do not explicitly address the issue, as discussed in Appendix C.3.

## 3.2 CREATE A SURROGATE FOR GLOBAL DATA

Because the major difference between global and local models is the data distribution that they should handle, we argue that explicitly seeking and learning from a surrogate for the global data is a more pertinent strategy for local updates. Regarding that the only source of global information in local updates is the downloaded global model, we try to solve the problem by augmenting local data

with the help of the global model. In this way, the augmented data can capture information beyond the local domain.

To fulfill this vision, we first adopt the augmentation network proposed in (Suzuki, 2022). This augmentation model, which consists of geometry and color augmentation modules, is fully optimizable via gradient descent (see Appendix D.2 for more details). Formally, the augmentation network is denoted as $a(\cdot; \phi) : \mathcal{X} \rightarrow \mathcal{X}$, where it takes an input image and outputs an augmented version, parameterized by $\phi$. We employ the following reduction objective to optimize $\phi_i$ in each local client:

$$\max_{\phi_i} \frac{1}{|\hat{D}_i|} \sum_{(x,y) \in \hat{D}_i} \left[ \ell(f(a(x; \phi_i); \theta_i + \Delta_i), y) - \ell(f(a(x; \phi_i); \theta), y) \right], \tag{9}$$

where $\Delta_i := \text{argmax}_\Delta \hat{\mathcal{E}}_D(\theta_i + \Delta)$ is introduced to facilitate theoretical proof. The objective above seeks to maximize the empirical risk for the local model, which functions as adversarial augmentation, supplementing the information not retained by the local model. Simultaneously, it minimizes the empirical risk for the global model, ensuring that the augmented images remain recognizable by the global model. By combining these two objectives, the augmented data serves as a meaningful surrogate for the global data, preserving global information during local training by alternately minimizing the risk on the augmented data. To validate this, we empirically demonstrate that the forgetting rate of the model trained on augmented data is lower than that of the model trained on local data (see Appendix C.2 for more details). It is important to note that $\{\phi_i\}$ are not designed to directly estimate the global data distribution in a static way. Instead, constrained by the global model, the augmented data is adversarially learned. For simplicity, we denote Eq. (9) as $\hat{\mathcal{E}}_{a(D_i; \phi_i)}(\theta_i + \Delta_i) - \hat{\mathcal{E}}_{a(D_i; \phi_i)}(\theta)$ (excluding the max operation). The resulting local objective with data augmentation is:

$$\min_{\theta_i} \max_{\|\Delta_a\| < \gamma} \frac{1}{|\hat{D}_i|} \sum_{(x,y) \in \hat{D}_i} \ell(f(a(x; \phi_i); \theta_i + \Delta_a), y)$$

$$\text{s.t. } \phi_i = \text{argmax}_{\phi_i} \left[ \hat{\mathcal{E}}_{a(D_i; \phi_i)}(\theta_i + \Delta_i) - \hat{\mathcal{E}}_{a(D_i; \phi_i)}(\theta) \right]. \tag{10}$$

To be noticed, Eq. (10) above has theoretical value: the risk on the augmented images is an upper bound of the risk on the global data. Assume the augmentation model is strong enough and denote the parameters of the augmentation model that augments local distribution into global distribution as $\hat{\phi}_i$ such that $a(D_i; \hat{\phi}_i) = D$. It is obvious that:

$$\hat{\mathcal{E}}_D^\gamma(\theta_i) = \hat{\mathcal{E}}_D(\theta_i + \Delta_i) = \hat{\mathcal{E}}_{a(D_i; \hat{\phi}_i)}(\theta_i + \Delta_i) \leq \max_{\phi_i} \hat{\mathcal{E}}_{a(D_i; \phi_i)}(\theta_i + \Delta_i). \tag{11}$$

Eq. (10) can be viewed as a practical substitute of $\max_{\phi_i} \hat{\mathcal{E}}_{a(D_i; \phi_i)}(\theta_i + \Delta_i)$ by restricting the augmented data to the range where they are recognizable by the global model. It avoids destructive adversarial augmentation with no limits. Thus, by assuming the inequality in the same form of Eq. (11) holds (which is easy to hold in practice when optimizing $\phi_i$):

$$\hat{\mathcal{E}}_D^\gamma(\theta_i) \leq \hat{\mathcal{E}}_{a(D_i; \phi_i)}(\theta_i + \Delta_i) \text{ s.t. } \phi_i = \text{argmax}_{\phi_i} \hat{\mathcal{E}}_{a(D_i; \phi_i)}(\theta_i + \Delta_i) - \hat{\mathcal{E}}_{a(D_i; \phi_i)}(\theta), \tag{12}$$

the generalization bound for the federated domain generalization task comes out as follows.

**Theorem 2.** *Denote the local models as $\{\theta_i\}_{i=1}^{M_S}$, the global model as $\theta$, and the augmentation models as $\{\phi_i\}_{i=1}^{M_S}$. Suppose $\{\theta_i\}_{i=1}^{M_S}$ satisfies Assumption 1, $\theta$ is the aggregate of $\{\theta_i\}_{i=1}^{M_S}$ and $p_i = 1/M_s$. For any $\theta \in \Theta$, the following bound holds with probability at least $1 - \delta$:*

$$\mathcal{E}_T(\theta) < \sum_i^{M_s} \frac{1}{M_s} \hat{\mathcal{E}}_{a(D_i; \phi_i)}^\gamma(\theta_i) + \frac{1}{2M_s} \sum_{i=1}^{M_s} \mathbf{Div}(D_i, T) + \max_{k \in [1, K]} \sqrt{\frac{v_k \ln(n/v_k) + \ln(K/\delta)}{n}}, \tag{13}$$

*where $\phi_i = \text{argmax}_{\phi_i} \hat{\mathcal{E}}_{a(D_i; \phi_i)}(\theta_i + \Delta_i) - \hat{\mathcal{E}}_{a(D_i; \phi_i)}(\theta)$.*

Theorem 2 shows that the risk on the test domain is upper bounded by the robust empirical risks of local clients on augmented samples, combined with the domain discrepancy, and a confidence bound. This implies that the performance on the unseen domain is directly related to the flatness of the seen clients on augmented samples. More discussions about the Theorem 2 and Theorem 1 can be found in Appendix B.1.

---

**Algorithm 1** Global Flat Minima

---

**Input:** global model $\theta = \theta^0$, $M_s$ seen clients models $\{\theta_i\}_{i=1}^{M_s}$ and datasets $\{D_i^s\}_{i=1}^{M_s}$, $R$ rounds, neighborhood radius $\gamma > 0$, local updates $E$, learning rate $\rho, \rho_\phi$, update interval $c$

**Output:** global model $\theta^R$

1: Initialize global model $\theta^0$, augmentation models $\{\phi_i\}_{i=1}^{M_s}$
2: **for** r=1,2,$\cdots$,R **do**
3:   **on client** $i$ **in parallel do**
4:   Initial local model $\theta_i^r = \theta^{r-1}$
5:   **for** e=1,2,$\cdots$,E **do**
6:     Sample a mini-batch $X_i$ from $D_i^s$
7:     Compute $\Delta = \gamma \nabla_\theta \hat{\mathcal{E}}_{a(D_i;\phi_i)}(\theta_i^r) / \|\nabla_\theta \hat{\mathcal{E}}_{a(D_i;\phi_i)}(\theta_i^r)\|_2$ on $X_i$     Inner maximization of $\theta_i$
8:     Compute $g_i = \nabla_\theta \hat{\mathcal{E}}_{a(D_i;\phi_i)}(\theta_i^r + \Delta)$                     Compute gradients on $\theta_i + \Delta$
9:     Update $\theta_i^r = \theta_i^r - \rho g_i$ on $X_i$
10:     **if** $e \% c == 0$ **then**
11:       Compute $g_{\phi_i} = \nabla_{\phi_i} \hat{\mathcal{E}}_{a(D_i;\phi_i)}(\theta_i) - \hat{\mathcal{E}}_{a(D_i;\phi_i)}(\theta)$ on $X_i$
12:       Update $\phi_i = \phi_i + \rho_\phi g_{\phi_i}$                     Update augmentation model $\phi_i$
13:     **end if**
14:   **end for**
15:   Update $\theta^r = \sum_i p_i \theta_i^r$
16: **end for**

---

### 3.3 OVERALL ALGORITHM

In this section, we present the practical and comprehensive algorithm of GFM. We begin by considering Eq. (10) in the local updates. From our experiments, the generalization performance is negligibly affected by the inclusion of the term $\Delta_i$. Both $\theta_i$ and $\theta_i + \Delta_i$ exhibit similar effects concerning augmentation; hence, we omit the plus operation to improve memory and computational efficiency. For the inner maximization $\max_{\|\Delta_a\| < \gamma}$, which aims to achieve flatness on augmented data, we employ the SAM optimizer proposed in (Foret et al., 2020). SAM serves as an optimizer for parameters of $\theta_i$ and the optimizing objective is of the min-max form:

$$\min_{\theta_i} \hat{\mathcal{E}}_{a(D_i;\phi_i)}^{\gamma}(\theta_i) \quad \text{and} \quad \max_{\phi_i} \left[ \hat{\mathcal{E}}_{a(D_i;\phi_i)}(\theta_i) - \hat{\mathcal{E}}_{a(D_i;\phi_i)}(\theta) \right]. \tag{14}$$

We solve this problem iteratively, optimizing $\theta_i$ and $\phi_i$ in alternating steps. The updates of $\theta_i$ and $\phi_i$ are adversarial, corresponding to Lines 5-12 in Algorithm 1. Specifically, $\phi_i$ is updated based on the maximization objective $\max_{\phi_i} \left[ \hat{\mathcal{E}}_{a(D_i;\phi_i)}(\theta_i) - \hat{\mathcal{E}}_{a(D_i;\phi_i)}(\theta) \right]$, while $\theta_i$ is updated based on the minimization $\min_{\theta_i} \hat{\mathcal{E}}_{a(D_i;\phi_i)}^{\gamma}(\theta_i)$. Proposition 1 provides the saddle point solution for the min-max process under certain simplifications. The min-max process will converge to the saddle point once the model reaches its neighborhood and will be stable. It can be inferred that the saddle point solution described in Proposition 1 is desirable because it achieves comparable global performance to the global model $\theta$, as demonstrated by $p(y|x;\theta_i^*) = s \cdot p(y|x;\theta)$. In this way, the local update can be effectively supplemented with global information as stated, leveraging both the global model and the augmentation model. The formal statement and further analysis can be found in Appendix F.

**Proposition 1.** *(Informal) Construct $\theta_i^*$ where $p(y|x;\theta_i^*) = s \cdot p(y|x;\theta)$ for any $x$ in the support set and its true label $y$. There exists $\phi_i^*$ such that $\theta_i^*$ is the local minimum of $\mathcal{E}_{a(D_i;\phi_i^*)}(\theta_i^*)$. Then, $(\theta_i^*, \phi_i^*)$ constitutes a saddle point solution of the min-max process.*

After local updates, the models are uploaded to the server and averaged following FedAvg. The averaged model is then distributed to each local client. The overall algorithm is shown in Algorithm 1.

## 4 EXPERIMENTS

### 4.1 EXPERIMENTAL SETTINGS

We use the following FedDG benchmarks to evaluate different methods: Digits-DG (Zhou et al., 2020) (24,000 images, 10 classes, four domains), PACS (Li et al., 2017) (9,991 images, 7 classes,

Table 1: Average classification accuracy using leave-one-domain-out validation. GFM (X) indicates the method combining GFM and X.

| Method | Digits-DG ConvNet | PACS ResNet18 | OfficeHome ResNet18 | TerraInc ResNet50 | Avg. |
|---|---|---|---|---|---|
| ***FL methods*** | | | | | |
| FedAvg (McMahan et al., 2017) | 67.46±0.27 | 82.56±0.47 | 64.82±0.28 | 44.23±0.69 | 64.77 |
| Scaffold (Karimireddy et al., 2020) | 68.30±0.79 | 82.54±0.25 | 64.56±0.20 | 42.70±0.46 | 64.53 |
| FedDyn (Acar et al., 2021) | 68.18±0.14 | 82.73±0.24 | 63.89±0.13 | 44.28±0.71 | 64.77 |
| MOON (Li et al., 2021) | 65.79±0.98 | 82.65±0.53 | 62.87±0.13 | 43.73±0.77 | 63.76 |
| FedSAM (Caldarola et al., 2022) | 66.67±0.49 | 83.36±0.22 | 65.28±0.35 | 45.16±1.36 | 65.12 |
| FedGAMMA (Dai et al., 2023) | 67.70±1.54 | 82.83±0.34 | 65.38±0.12 | 43.56±1.04 | 64.87 |
| FedSMOO (Sun et al., 2023) | 69.43±0.53 | 82.92±0.79 | 62.40±0.22 | 43.38±0.76 | 64.53 |
| ***FedDG methods*** | | | | | |
| FedSR (Nguyen et al., 2022) | 68.21±0.38 | 83.20±0.83 | 63.99±0.31 | 42.97±0.93 | 64.59 |
| GA (Zhang et al., 2023a) | 68.45±0.16 | 83.39±0.61 | 65.11±0.05 | 45.59±0.98 | 65.64 |
| StableFDG (Park et al., 2024) | 67.80±0.89 | 84.22±0.72 | 64.61±0.02 | 44.48±0.14 | 65.28 |
| FedIIR (Guo et al., 2023b) | 69.25±0.25 | 83.94±0.16 | 60.64±0.33 | 46.88±0.80 | 65.18 |
| *GFM* | *69.72±0.99* | 84.46±0.42 | *65.57±0.19* | 46.02±1.04 | *66.44* |
| GFM (GA) | **71.32±0.64** | **84.97±0.22** | **66.08±0.20** | *46.91±0.54* | **67.32** |
| GFM (FedIIR) | 69.57±1.12 | *84.67±0.40* | 61.74±0.36 | **47.66±0.82** | 65.91 |

four domains), OfficeHome (Venkateswara et al., 2017) (15,588 images, 65 classes, four domains), and TerraInc (Beery et al., 2018) (24,788 images, 10 classes, four domains). These benchmarks were selected to cover a broad range of conditions in digital and real-world scenarios. Leave-one-domain-out evaluation is carried out for all benchmarks, which by turn keeps data from one domain as the unseen client for testing and distributes each other domain data to a training client. The backbone architectures used are a CNN proposed in (Zhou et al., 2020) for Digits-DG, ImageNet-pretrained ResNet18 (He et al., 2016) for PACS and OfficeHome benchmarks, and ImageNet-pretrained ResNet50 (He et al., 2016) for the TerraInc benchmark. For the SAM optimizer, $\gamma$ is set as 0.02. More details can be found in Appendix D.

## 4.2 FedDG performance

Two components of GFM are the novel augmentation strategy and the approach (SAM optimizer in our experiments) minimizing the robust risk locally. Because it is orthogonal to some previous works, we show the superior performance of GFM in two ways: 1) direct comparisons with previous FedDG and federated learning (FL) baselines; and 2) combining GFM with other approaches. The considered baselines are briefly introduced as follows:

**FedAvg (McMahan et al., 2017):** the commonly used baseline for the Federated learning.

**Scaffold (Karimireddy et al., 2020):** utilized variance reduction techniques to correct client drift.

**FedDyn (Acar et al., 2021):** incorporated dynamic regularization to improve convergence.

**MOON (Li et al., 2021):** applied contrastive learning between global and local models.

**FedSAM (Caldarola et al., 2022):** adopted SAM optimizer in local client training.

**FedGAMMA (Dai et al., 2023):** introduced a gradient matching mechanism with SAM optimizer.

**FedSMOO (Sun et al., 2023):** enforced high consistency in local SAM perturbations by ADMM.

**FedSR (Nguyen et al., 2022):** aimed to learn a simple data representation for better generalization.

**GA (Zhang et al., 2023a):** aggregated models in the server according to generalization gaps.

**StableFDG (Park et al., 2024):** enabled each client to explore novel styles by style sharing.

**FedIIR (Guo et al., 2023b):** aligned the gradients of different clients to derive an invariant classifier. FedIIR and FedIIR (GFM) are not directly comparable to other baselines. (Appendix D.1)

Tab. 1 gives the summarized results of experiments with different methods, while detailed results of each single test domain are given in Tab. 3 (Digit-DG and PACS) and Tab. 4 (OfficeHome and TerraInc) in Appendix C.1. From these tables, we can conclude that GFM only (GFM + FedAvg) can achieve SOTA performance on average and on many datasets. What's more, the direct comparisons between FedSAM and GFM indicate the need beyond local flatness for FedDG, demonstrating the effectiveness of the proposed global model-constrained adversarial data augmentation. Further, combining GFM with other methods can consistently improve the generalization ability and achieve

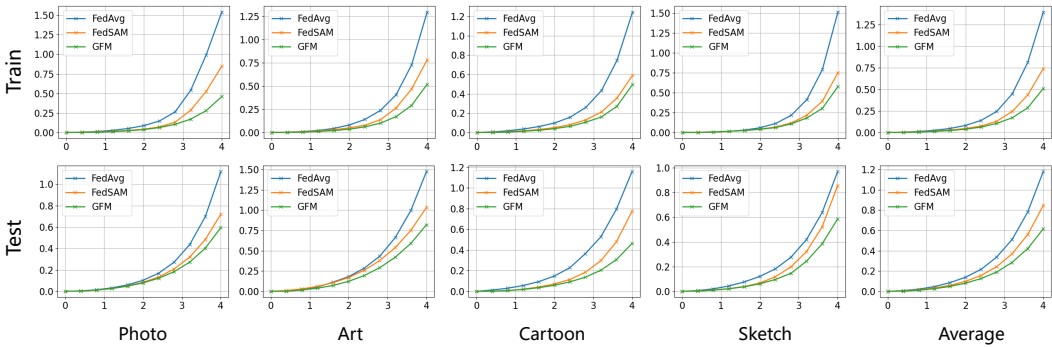

Figure 1: Quantitative results of flatness measured by $F_\gamma(\theta)$. Each column represents an independent experiment. (For example, the first column represents the experiment with the photo domain as the unseen test client in the leave-one-domain-out evaluation setting.) The train results are calculated on data of all seen clients, while the test results are on the unseen test domain. For each figure, the Y-axis indicates the flatness $F_\gamma(\theta)$ and the X-axis indicates the radius $\gamma$.

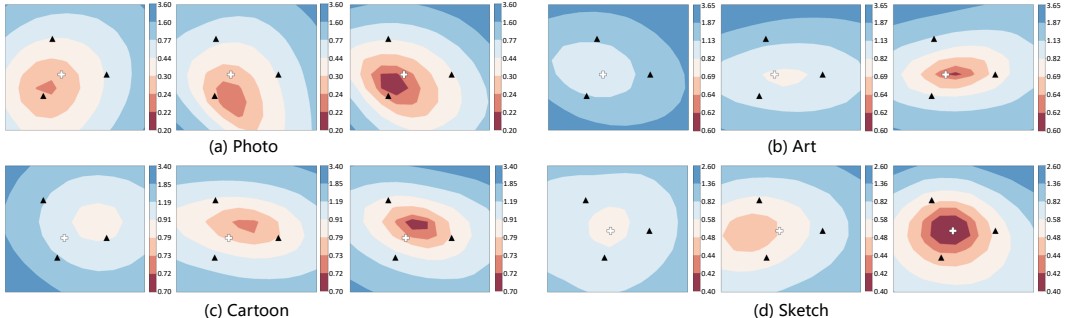

Figure 2: Test loss surface visualization on PACS. In each subfigure, from left to right, the contours belong to FedAvg, FedSAM, and GFM respectively. Triangle marks indicate local models and cross marks indicate the global model. The color bars are log-normalized and one can approximately compare flatness by observing the size of regions at or above the third level (high to low). We use a similar visualization technique as in (Garipov et al., 2018).

better performance, especially for GFM (GA). GFM (GA) surpasses the previous SOTA method by 1.7 percent on average. The success of GFM (GA) can be attributed to improved flatness in both the local training and the aggregation stage.

## 4.3 FLATNESS COMPARISONS

In this section, we empirically compare the flatness of solutions found by GFM and other methods. Specifically, we use expected loss value changes $F_\gamma(\theta)$ proposed in (Cha et al., 2021) as a metric. For model with parameter $\theta$, $F_\gamma(\theta)$ calculates the expected loss changes between $\theta$ and $\theta + \gamma$ on the sphere of radius $\gamma$ as follows:

$$F_\gamma(\theta) := \mathop{\mathbb{E}}_{||\theta'||=||\theta||+\gamma} [\mathcal{E}(\theta') - \mathcal{E}(\theta)]. \tag{15}$$

Large $F_\gamma(\theta)$ indicates the loss changes dramatically when moving from $\theta$ to the sphere of radius $\gamma$, which reveals a sharp minimum and vice versa. One can effectively estimate $F_\gamma(\theta)$ with finite samples according to the Monte-Carlo method, because $F_\gamma(\theta)$ has an unbiased finite sample estimator and is computationally efficient. In our experiments, $F_\gamma(\theta)$ is approximated with 50 samples. We quantitatively measure $F_\gamma(\theta)$ of the global model trained by FedAvg, FedSAM, and GFM with all unseen domains of the PACS dataset. FedAvg represents the baseline without a special design for flatness, FedSAM focuses on the flatness of local models, while the proposed GFM tries to approach global flatness. The results are reported in Fig. 1. We can conclude from it that both FedSAM and GFM can help improve global flatness, and GFM can find flatter minima than both FedSAM and FedAvg in all experiments and on both the seen train datasets and the unseen test dataset, which verifies the effectiveness of GFM for seeking global flat minima.

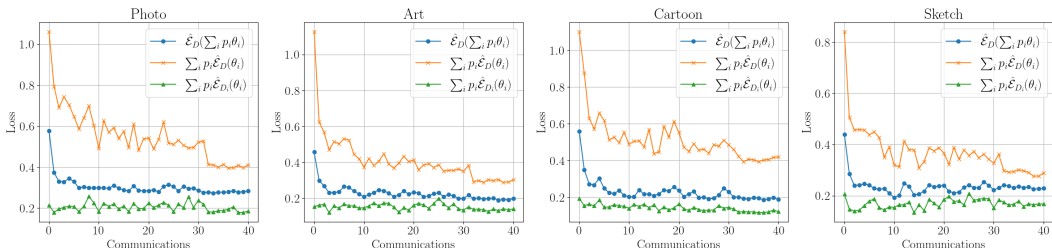

Figure 3: Empirical validation of Assumption 1 in the training stage on the PACS dataset.

Besides random directions measured by $F_\gamma(\theta)$, we also consider the special cases within the aggregation plane. We plot the test loss surfaces of three local models and the aggregated global model derived from different methods on the PACS dataset in Fig. 2. In a similar vein as (Caldarola et al., 2022), models of clients in FedAvg are positioned in relatively high-loss regions and thus the resulting global model is far away from a good minimum. Fortunately, seeking flatter minima in local updates can ameliorate the situation and tend to find solutions in flatter and low-loss regions. Results in Fig. 2 suggest that solutions of GFM on Art and Cartoon test domain meet this expectation strictly while these on Photo and Sketch meet it partially by finding solutions in low-loss areas with comparable flatness. What's more, the loss surfaces of FedSAM can be viewed as the "middle point" between FedAvg and GFM.

### 4.4 EMPIRICAL VALIDATION OF ASSUMPTION 1

The essential premise for Theorem 2 to hold is the validity of Assumption 1. This section empirically examines if Assumption 1 holds. For better illustration, we compare risks calculated in three different ways in the federated learning setting: $\hat{\mathcal{E}}_D(\sum_i p_i\theta_i)$, $\sum_i p_i\hat{\mathcal{E}}_D(\theta_i)$, and $\sum_i p_i\hat{\mathcal{E}}_{D_i}(\theta_i)$. $\sum_i p_i\hat{\mathcal{E}}_{D_i}(\theta_i)$ can be viewed as the lower bound of $\hat{\mathcal{E}}_D(\sum_i p_i\theta_i)$ and $\sum_i p_i\hat{\mathcal{E}}_D(\theta_i)$ because it averages the risk of the optimal model in each client. As for the other two terms, we assume $\hat{\mathcal{E}}_D(\sum_i p_i\theta_i) \leq \sum_i p_i\hat{\mathcal{E}}_D(\theta_i)$ in Assumption 1, which is a natural assumption to make the aggregation meaningful. The empirical results on the PACS dataset are given in Fig. 3. From it, we can conclude that Assumption 1 holds empirically in every communication round for every test domain.

### 4.5 ABLATION STUDY

The core intuition behind the overall algorithm is to approach global flatness. In this section, we aim to investigate the impact of improved flatness on the global model. This analysis is challenging because global flatness is highly coupled with the proposed global model-

Table 2: Ablation Study

| Method | Digits | PACS | OfficeHome | TerraInc | Avg. |
|--------|--------|-------|------------|----------|-------|
| FedAvg | 67.46 | 82.56 | 64.82 | 44.23 | 64.77 |
| FedSAM | 66.67 | 83.36 | 65.28 | 45.16 | 65.12 |
| GCA | 69.65 | 83.91 | 64.64 | 44.20 | 64.93 |
| GFM | **69.72** | **84.46** | **65.57** | **46.02** | **66.44** |

constrained adversarial data augmentation (GCA) component. Furthermore, as an augmentation strategy, GCA can independently influence generalization performance. To address this, we perform extensive ablation studies on GFM across a wide range of datasets to uncover insights into global flatness.

The components of GFM include the global model-constrained adversarial data augmentation strategy (GCA) and the SAM optimizer. Both GCA and SAM are applied during the local training stage and can be used independently. This results in four possible combinations: (1) FedAvg: GFM reduces to the FedAvg baseline without the SAM optimizer and GCA. (2) FedSAM: GFM reduces to the FedSAM baseline without GCA. (3) GCA: The FedAvg baseline enhanced with GCA. (4) GFM: The complete method, incorporating both GCA and the SAM optimizer.

From the results in Table 2, we observe that FedSAM, leveraging local flatness, achieves improved generalization performance in three cases, while GCA demonstrates significant effectiveness on the Digits and PACS datasets. By combining the benefits of improved global flatness and the effective data augmentation strategy, GFM achieves the best performance across four datasets. Notably, in

the OfficeHome and TerraInc datasets, GCA alone does not enhance generalization performance, which underscores the importance and effectiveness of the stated global flatness.

### 4.6 PARAMETER ANALYSIS

In this section, we demonstrate the selection of hyperparameters. There are two key hyperparameters in GFM: the radius $\gamma$ and the update interval $c$. The radius $\gamma$ is a critical hyperparameter in SAM-based methods, as it determines the range of model perturbation. The optimal value of $\gamma$ varies across tasks, datasets, and models. In our experiments, we conducted a grid search for $\gamma$ on the PACS dataset to determine the appropriate value for both FedSAM and GFM. For GFM,

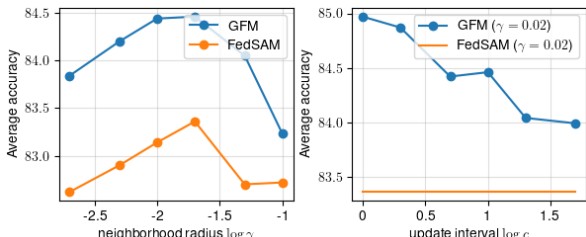

Figure 4: Influences of radius $\gamma$ and update interval $c$. The values are presented in logarithmic scale.

the update interval $c$ is fixed to 10. The results are shown on the left side of Figure 4. The accuracy first increases and then decreases as $\gamma$ increases, indicating the existence of a local optimum. This behavior is expected because, with a small $\gamma$, FedSAM recovers to the FedAvg baseline (and GFM reverts to the FedAvg+GCA baseline), resulting in reduced performance. Conversely, when $\gamma$ is too large, the SAM optimizer becomes unstable and struggles to converge. Notably, GFM exhibits a relatively flatter optimum compared to FedSAM. This could be attributed to the improved consistency of local models in GFM, which reduces the need for local flatness to achieve sufficient global flatness. In our experiments, we found that $\gamma = 0.02$ achieves the optimal performance for both methods. Therefore, we set $\gamma = 0.02$ for subsequent experiments on PACS and other datasets.

With $\gamma$ fixed, we tune the update interval $c$, which controls the update frequency of the augmentation model. With a larger value of $c$, the augmentation model tends to update less frequently with a relatively low computational cost. As shown on the right side of Figure 4, the performance improves with more frequent updates of the augmentation model. The strategy of alternating one iteration of augmentation with one iteration of classification achieves the best performance. However, this approach incurs a significantly higher computational cost, as illustrated in Appendix C.5. To balance performance and efficiency, we set $c = 10$ for related experiments.

## 5 LIMITATIONS

One limitation of GFM is the increased computational cost for local updates. The inclusion of the augmentation method and the SAM optimizer in the local client results in higher computational demands compared to the baseline method. Details on the exact computational overhead and potential trade-offs can be found in Appendix C.5. Another potential limitation of our current approach is the restriction in the types of augmentation transformations. At present, the augmentation model is limited to applying color and geometry augmentations. However, other forms of augmentation, such as Fourier-based transformations, could also be beneficial for domain generalization (DG). Identifying and exploring additional augmentation techniques, or even leveraging generative models, represents a promising avenue for future research.

## 6 CONCLUSION

In this paper, we propose a novel algorithm, named GFM, to seek global flat minima in FedDG. The overall algorithm is explainable by viewing it as minimizing the upper bound of the robust risk of the global model on the global data distribution. Specifically, we propose the global model-constrained adversarial data augmentation strategy to seek a surrogate for global data and use sharpness-aware minimization to pursue flatter minima. Flatness measurement and loss surface visualization experiments validate the flatter minima of the global model found by GFM than by FedAvg and the method seeking local flatness. Furthermore, extensive experiments on four FedDG benchmarks confirmed the improved performance of GFM when comparing or combining with previous works.

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

## A    RELATED WORK

**Domain generalization**    Domain generalization aims to address the distribution shift problem caused by domain gaps and enable the model to perform well not only on train domains but also on test domains. Some researchers focus on the domain alignment idea (Li et al., 2018b;c; Shao et al., 2019; Shi et al., 2021b) by bridging domain distribution gaps. There are also some works (Balaji et al., 2018; Zhao et al., 2021; Li et al., 2019) considering meta-learning strategies to learn from domain shifts. Other techniques including invariant risk minimization (Ahuja et al., 2020; Arjovsky et al., 2019; Krueger et al., 2021), data augmentations (Huang et al., 2021; Xu et al., 2021; Zhang et al., 2022), and self-supervised learning (Carlucci et al., 2019; Kim et al., 2021; Wang et al., 2020) are also validated effectively in domain generalization. However, these centralized methods either require domain labels or need data samples from all domains, which is not achievable in the federated learning setting due to the data privacy issue.

**Federated domain generalization**    Federated domain generalization involves both domain generalization and federated learning, which aims to bridge the participating gap of unseen clients with domain shifts. Methods with different motivations are proposed to solve it. FedADG (Zhang et al., 2023a) aligned each seen client's data representation distribution by adversarial training for getting universal representation, FedSR (Nguyen et al., 2022) tried to learn simple representation for avoiding spurious correlation by regularizing the feature norm and conditional mutual information, FedIIR (Guo et al., 2023b) implicitly learned invariant classifier by gradient alignment, GA (Zhang et al., 2023b) focused on the averaging stage and adjusted coefficients of local models by their performance, and StableFDG (Park et al., 2024) and CCST (Chen et al., 2023) proposed to utilize style statistics in seen clients to help local training. Different from them, we try to approach the global flatness for improved domain generalization ability.

**Flat minima**    A popular perspective on the generalization of deep networks is that flat minima are robust to test distribution shifts. This problem, explored early (Hinton & Van Camp, 1993; Hochreiter & Schmidhuber, 1994; 1997), has seen a resurgence in recent years (Dziugaite & Roy, 2017; Li et al., 2018a; Jiang et al., 2019), showing a strong relation between flat minima and generalization. Recent works seek flat minima either during optimization (Foret et al., 2020; Kwon et al., 2021; Zhuang et al., 2022; Sun et al., 2024; Zhong et al., 2022) or via post-processing (Izmailov et al., 2018; Cha et al., 2021). The former is exemplified by Sharpness-Aware Minimization (SAM) (Foret et al., 2020), which minimized robust risk, while later works (Kwon et al., 2021; Zhuang et al., 2022; Sun et al., 2024; Zhong et al., 2022) overcame the shortcomings of SAM or proposed new theoretical explanations. Wen et al. (2024) comprehensively discussed the relationship between flatness, generalization, and SAM with respect to different architectures and data distributions. Post-processing methods, such as SWA (Izmailov et al., 2018), exploited linear mode connectivity (Draxler et al., 2018; Garipov et al., 2018; Juneja et al., 2022) by averaging models along SGD paths to improve generalization. In this paper, we focus on the flatness of the global model in federated learning, which is not directly optimizable.

## B    DISCUSSIONS

### B.1    MORE DISCUSSIONS ABOUT THEOREMS

The distinction between Theorem 1 and Theorem 2 is in the first term on the RHS of Eq. (4) and Eq. (13). To minimize the objective $\mathcal{E}_T(\theta)$, Theorem 1 suggests minimizing $\hat{\mathcal{E}}^\gamma_{\mathcal{D}^s}(\theta)$, which is straightforward in a centralized setting where data from each $D^s_i$ is aggregated, and $\Delta$ can be estimated as $\Delta = \gamma \frac{\nabla \hat{\mathcal{E}}_{\mathcal{D}^s}(\theta)}{|\nabla \hat{\mathcal{E}}_{\mathcal{D}^s}(\theta)|}$. However, in the federated learning scenario, where data remains private, the global gradient $\nabla \hat{\mathcal{E}}_{\mathcal{D}^s}(\theta)$ is inaccessible, and therefore the global $\Delta$ cannot be easily estimated. This prevents clients from cooperating to minimize $\hat{\mathcal{E}}^\gamma_{\mathcal{D}^s}(\theta)$, rendering Theorem 1 inapplicable in federated settings. In contrast, Theorem 2 suggests minimizing $\sum_{i=1}^{M_s} \frac{1}{M_s} \hat{\mathcal{E}}^\gamma_{a(D_i;\phi_i)}(\theta_i)$. Here, $\Delta$ (denoted as $\Delta_i$ with slight abuse of notation) for each local model $\theta_i$ can be estimated locally on $a(D_i; \phi_i)$ by $\Delta = \gamma \frac{\nabla \hat{\mathcal{E}}_{a(D_i;\phi_i)}(\theta_i)}{|\nabla \hat{\mathcal{E}}_{a(D_i;\phi_i)}(\theta_i)|}$. The location of the maximization changes, and this

computation is performed locally during the clients' updates. It is worth noting that the introduced data-centric bounding inequality can not only be applied to Eq. (4) but also to other empirical risk bounds, such as Theorem 1 in DomainDrop (Guo et al., 2023a).

## C ADDITIONAL EXPERIMENTAL RESULTS

### C.1 RESULTS FOR EACH DOMAIN

Tab. 3 and Tab. 4 provide detailed results for each domain on Digits-DG, PACS, OfficeHome, and TerraInc dataset.

Table 3: Results on Digits-DG and PACS. GFM(X) indicates the method combining GFM and X.

| Method | Digits-DG | | | | | PACS | | | | |
|---|---|---|---|---|---|---|---|---|---|---|
| | MNIST | MNIST-M | SVHN | SYN | Avg. | Art | Cartoon | Photo | Sketch | Avg. |
| FedAvg | 90.67 | 44.98 | 50.16 | 84.04 | 67.46 | 77.41 | 77.82 | 92.67 | 82.35 | 82.56 |
| Scaffold | 90.60 | 45.91 | 52.50 | 84.18 | 68.30 | 77.73 | 78.00 | 92.73 | 81.69 | 82.54 |
| FedDyn | 89.88 | 46.16 | 52.15 | 84.53 | 68.18 | 78.53 | 78.67 | 92.75 | 80.95 | 82.73 |
| MOON | 90.33 | 42.79 | 46.09 | 83.95 | 65.79 | 77.90 | 76.88 | 93.29 | 82.51 | 82.65 |
| FedSAM | 92.27 | 44.33 | 47.05 | 83.02 | 66.67 | 78.34 | 78.85 | 92.45 | 83.81 | 83.36 |
| FedGAMMA | 92.27 | 44.50 | 50.08 | 83.93 | 67.70 | 77.80 | 78.10 | 92.52 | 82.99 | 82.83 |
| FedSMOO | 90.29 | 44.38 | **57.90** | 85.15 | 69.43 | 78.77 | 77.49 | 90.84 | 84.58 | 82.92 |
| FedSR | 92.84 | 48.17 | 46.15 | 85.69 | 68.21 | 81.95 | 74.37 | 92.93 | 81.41 | 82.67 |
| GA | 91.34 | 44.53 | 53.24 | 84.70 | 68.45 | 80.93 | 77.30 | 94.49 | 80.84 | 83.39 |
| FedSDG | 88.56 | 49.34 | 51.18 | 82.11 | 67.80 | 81.61 | 78.81 | **94.71** | 81.76 | 84.22 |
| FedIIR | 92.28 | **49.95** | 51.30 | 83.46 | 69.25 | 82.13 | 77.27 | 93.91 | 82.46 | 83.94 |
| GFM | 92.22 | 46.28 | 56.27 | 84.12 | 69.72 | 80.34 | 78.27 | 92.53 | **86.70** | 84.46 |
| GFM (GA) | **93.37** | 48.21 | 57.69 | **85.99** | **71.32** | **82.96** | 76.92 | 93.99 | 86.00 | **84.97** |
| GFM (FedIIR) | 91.16 | 49.45 | 54.62 | 83.05 | 69.57 | 81.58 | **79.03** | 93.73 | 84.33 | 84.67 |

Table 4: Results on OfficeHome and TerraInc. GFM(X) indicates the method combining GFM and X.

| Method | OfficeHome | | | | | TerraInc | | | | |
|---|---|---|---|---|---|---|---|---|---|---|
| | Art | Clipart | Product | Real | Avg. | L100 | L38 | L43 | L46 | Avg. |
| FedAvg | 57.88 | 53.45 | 73.65 | 74.28 | 64.82 | 53.03 | **41.64** | 46.05 | 36.18 | 44.23 |
| Scaffold | 56.87 | 53.84 | 73.51 | 74.03 | 64.56 | 51.76 | 40.91 | 42.65 | 35.49 | 42.70 |
| FedDyn | 56.94 | 52.73 | 72.55 | 73.32 | 63.89 | 51.82 | 40.55 | 46.29 | 38.44 | 44.28 |
| MOON | 55.45 | 51.90 | 71.72 | 72.42 | 62.87 | 51.00 | 43.29 | 44.48 | 36.16 | 43.73 |
| FedSAM | 57.13 | 55.46 | 74.39 | 74.14 | 65.28 | 55.26 | 40.83 | 46.13 | 38.42 | 45.16 |
| FedGAMMA | 57.34 | 55.10 | 74.58 | 74.51 | 65.38 | 53.70 | 37.65 | 46.26 | 36.53 | 43.56 |
| FedSMOO | 52.59 | 55.68 | 69.84 | 71.47 | 62.40 | 58.12 | 33.24 | 45.63 | 36.51 | 43.38 |
| FedSR | 56.40 | 53.94 | 72.07 | 73.55 | 63.99 | 50.22 | 38.99 | 44.11 | 38.55 | 42.97 |
| GA | 58.57 | 53.55 | 73.73 | 74.59 | 65.11 | 54.48 | 39.13 | 48.87 | 39.88 | 45.59 |
| FedSDG | 55.57 | **59.03** | 71.59 | 72.25 | 64.61 | **67.34** | 36.63 | 38.08 | 35.87 | 44.48 |
| FedIIR | 52.33 | 49.66 | 69.50 | 71.06 | 60.64 | 54.88 | 40.64 | 53.23 | 38.74 | 46.88 |
| GFM | 57.76 | 55.23 | **74.73** | 74.57 | 65.57 | 59.29 | 40.51 | 48.31 | 35.92 | 46.02 |
| GFM (GA) | **58.58** | 56.04 | 74.60 | **75.10** | **66.08** | 57.07 | 40.16 | 50.45 | **39.94** | 46.91 |
| GFM (FedIIR) | 54.30 | 51.35 | 69.49 | 71.84 | 61.74 | 60.02 | 38.75 | **54.16** | 37.70 | **47.66** |

### C.2 IS AUGMENTED DATA A BETTER SURROGATE FOR GLOBAL DATA THAN LOCAL DATA?

To answer this question properly, we focus on the trained model after local updates. Proposition 1 suggests that the trained model performs similarly to the global model when converging to the saddle point. Consequently, we evaluate the effect of augmented data and local data by measuring the mean forgetting rate between the trained model and the global model on the global dataset $\hat{D}$. The mean forgetting rate is defined as:

$$\bar{R}_f = \frac{1}{M_s} \sum_{i=1}^{M_s} \frac{\text{ACC}(\theta; \hat{D}) - \text{ACC}(\theta_i; \hat{D})}{\text{ACC}(\theta; \hat{D})}, \tag{A.1}$$

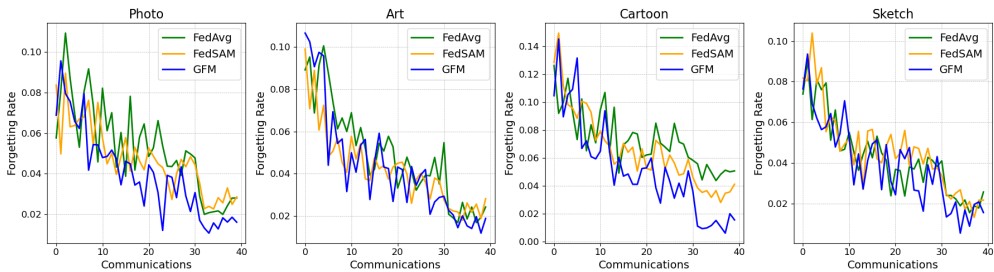

Figure 5: Forgetting rate of different methods during the federated training process on PACS.

where ACC denotes the classification accuracy function. A smaller $\bar{R}_f$ indicates that the model after local updates keeps more knowledge of the global data distribution, which is a desirable property.

We conduct experiments on the PACS benchmark and present the forgetting rate throughout the entire training process, as illustrated in Figure 5. Although $\bar{R}_f$ exhibits significant fluctuations during the training process, it is evident that GCA adopted in GFM effectively ameliorates the situation of forgetting, particularly in the later stages of training when the global model becomes stronger. These experiments provide a direct evaluation, demonstrating that augmented data serves as a better surrogate for global data compared to local data.

## C.3 RESULTS FOR REGULARIZATION-BASED METHODS

Given that the updates in local data distribution will incur catastrophic forgetting of global knowledge, one can pursue the enhancement of consistency between global and local models by knowledge distillation or penalizing model changes. In the meantime, we try to optimize local models to a flat region. As a result, the objective thus should combine the local training term and anti-forgetting term as follows:

Table 5: Results of example instantiation of Eq. (A.2).

| Method | Art | Cartoon | Photo | Sketch | Avg. |
|---|---|---|---|---|---|
| FedAvg | 77.41 | 77.82 | 92.67 | 82.35 | 82.56 |
| SAM | 78.74 | 79.42 | 92.61 | 83.17 | 83.49 |
| LWF | 79.74 | 78.65 | 93.87 | 80.90 | 83.29 |
| SAM+LWF | 79.26 | 79.05 | 92.69 | 83.79 | 83.70 |

$$\min_{\theta_i} \hat{\mathcal{E}}_{D_i}(\theta_i) + \mathcal{L}_{con}^{\gamma}(\theta, \theta_i), \tag{A.2}$$

where $\mathcal{L}_{con}$ measures the consistency between the global and local model. Some loss terms in previous works can be viewed as a special case of Eq. (A.2), such as the dynamic regularization term in (Sun et al., 2023). Here, we propose a simple baseline by adopting knowledge distillation (Li & Hoiem, 2017) loss as $\mathcal{L}_{con}$ and using SAM (Foret et al., 2020) optimizer. Results in Tab. 5 validate the effectiveness of Eq. (A.2). Compared to the proposed method in this paper, regularization-based methods can't explicitly minimize Eq. (8) and achieve inferior performance.

## C.4 IS GCA COMPATIBLE WITH OTHER AUGMENTATION?

Though designed for supplementing global information in the local training stage, GCA has similar forms (i.e., color transformation and geometric transformation) with other data augmentation strategies. So if GCA is compatible with other augmentation strategies remains unsolved. To figure it out, we combine GCA with three popular data augmentation methods in classification tasks: RandAug-

Table 6: Results when combining GCA with other popular augmentation strategies on the PACS dataset.

| Method | Art | Cartoon | Photo | Sketch | Average |
|---|---|---|---|---|---|
| RA | 83.04 | 78.46 | 93.83 | 83.50 | 84.71 |
| +GCA | **83.40** | **78.83** | **94.17** | 86.28 | **85.67** |
| AA | 82.57 | 77.43 | 93.91 | 84.47 | 84.60 |
| +GCA | 81.64 | 78.77 | 93.43 | **87.44** | 85.32 |
| Cutout | 76.98 | 77.77 | 92.12 | 80.74 | 81.90 |
| +GCA | 79.91 | 77.80 | 91.62 | 84.48 | 83.45 |

ment (RA) Cubuk et al. (2020), AutoAugment (AA) Cubuk et al. (2019), and Cutout DeVries & Taylor (2017). We conduct experiments on the PACS dataset and the results on PACS are shown in Tab. 6. We can draw some conclusions from it: (1) some popular augmentation strategies like RA and AA do a favor for the model's generalization performance while others (Cutout) don't;

Table 7: Computation overhead comparison between GFM and FedAvg.

| | Digits-DG | | PACS | | OfficeHome | | TerraInc | |
|---|---|---|---|---|---|---|---|---|
| | time | space | time | space | time | space | time | space |
| FedAvg | $V$ | $M, M$ | $V$ | $M, M$ | $V$ | $M, M$ | $V$ | $M, M$ |
| Scaffold | $\approx V$ | $4M, 2M$ | $\approx V$ | $4M, 2M$ | $\approx V$ | $4M, 2M$ | $\approx V$ | $4M, 2M$ |
| FedDyn | $\approx V$ | $3M, M$ | $1.34V$ | $3M, M$ | $\approx V$ | $3M, M$ | $\approx V$ | $3M, M$ |
| MOON | $\approx V$ | $3M, M$ | $1.38V$ | $3M, M$ | $\approx V$ | $3M, M$ | $\approx V$ | $3M, M$ |
| FedSAM | $1.08V$ | $M, M$ | $1.73V$ | $M, M$ | $1.43V$ | $M, M$ | $1.35V$ | $M, M$ |
| FedGAMMA | $1.09V$ | $4M, 2M$ | $1.70V$ | $4M, 2M$ | $1.51V$ | $4M, 2M$ | $1.32V$ | $4M, 2M$ |
| FedSMOO | $1.22V$ | $5M, 2M$ | $2.00V$ | $5M, 2M$ | $1.72V$ | $5M, 2M$ | $1.48V$ | $5M, 2M$ |
| FedSR | $\approx V$ | $M, M$ | $\approx V$ | $M, M$ | $\approx V$ | $M, M$ | $\approx V$ | $M, M$ |
| GA | $\approx V$ | $M, M$ | $\approx V$ | $M, M$ | $\approx V$ | $M, M$ | $\approx V$ | $M, M$ |
| GFM | $1.42V$ | $25.91M, M$ | $2.86V$ | $2.25M, M$ | $2.01V$ | $2.25M, M$ | $1.55V$ | $2.12M, M$ |
| GFM (GA) | $1.45V$ | $25.91M, M$ | $2.92V$ | $2.25M, M$ | $1.95V$ | $2.25M, M$ | $1.62V$ | $2.12M, M$ |

(2) further combining popular augmentation with GCA can still gain a non-trivial improvement. It indicates that though in similar forms, GCA can inject useful global information to the augmented data to achieve consistent performance gains. As a result, GCA is compatible and can be used with other data augmentation strategies practically.

## C.5 COMPUTATION OVERHEAD AND POTENTIAL TRADE-OFFS

In GFM, an augmentation model is incorporated into local updates to better approach a flat optimization landscape, albeit with some additional computational overhead. We compare the time and space overheads of GFM with other baseline methods. The space overhead is evaluated in two aspects: the parameters used during local updates (former element in the space column) and the parameters required for communication (latter element in the space column). From the results, it can be observed that the additional parameters introduced by the augmentation model are relatively small ($0.25M$ for ResNet-18 and $0.12M$ for ResNet-50), except for the small ConvNet architecture. Furthermore, the actual running time increases to approximately 1.5 to 1.7 times that of the FedSAM baseline, with variations due to differences in data processing times. Notably, among methods that aim to achieve flatter minima of the global model (including FedGAMMA, FedSMOO, and GFM), our method imposes the lowest space constraints.

The increase in running time is influenced by three factors: (1) the forward pass of the augmentation model to generate augmented data, (2) the backward pass when optimizing the augmentation model, and (3) the inner maximization within the SAM optimizer. Here, we focus on the first two components introduced by the augmentation model. One potential trade-off to reduce computational cost is to decrease the update frequency of the augmentation model. As shown in Tab. 8, reducing the frequency of updates for $\phi$ can lower the time cost by reducing the number of backward passes. Another strategy is to use low-dimensional images as inputs to the deep augmentation modules $c(; \phi_c)$ and $g(; \phi_g)$ (see Appendix D.2 for further clarification). Specifically, let $\bar{x}_i$ be the low-dimensional (e.g., 32x32) version of $x_i$. The color and geometry parameters, $\alpha_i, \beta_i = c(\bar{x}_i; \phi_c)$ and $A = g(\bar{x}_i; \phi_g)$, can be obtained from $\bar{x}_i$ and then applied to the original images ($\alpha_i, \beta_i$ need upsampling to the original dimension). This approach reduces the computational cost associated with the deep model during both the forward and backward passes. As shown in Table 9, dimension of inputs to the augmentation module have minimal impact on the final performance. By utilizing low-dimensional images as inputs, the computational cost of augmentation in GFM is reduced to approximately 30% of the FedSAM baseline.

Table 8: Performance for different update intervals $c$.

| Update Interval $c$ | 50 | 20 | 10 | 5 | 2 | 1 |
|---|---|---|---|---|---|---|
| Time | $2.34V$ | $2.66V$ | $2.86V$ | $3.23V$ | $4.51V$ | $6.94V$ |
| ACC | 83.99 | 84.04 | 84.46 | 84.42 | 84.87 | 84.97 |

## C.6 VISUALIZATION OF AUGMENTED IMAGES

In this section, we present visualizations of the augmented images from the PACS dataset. As discussed in Appendix D.2, the augmentation model comprises only geometry and color augmentation

Table 9: Impact of dimensions of inputs to the augmentation module.

| Image Dimension | $28 \times 28$ | $56 \times 56$ | $112 \times 112$ | $224 \times 224$ |
|---|---|---|---|---|
| Time | $2.30V$ | $2.31V$ | $2.44V$ | $2.86V$ |
| ACC | 84.35 | 84.30 | 84.37 | 84.46 |

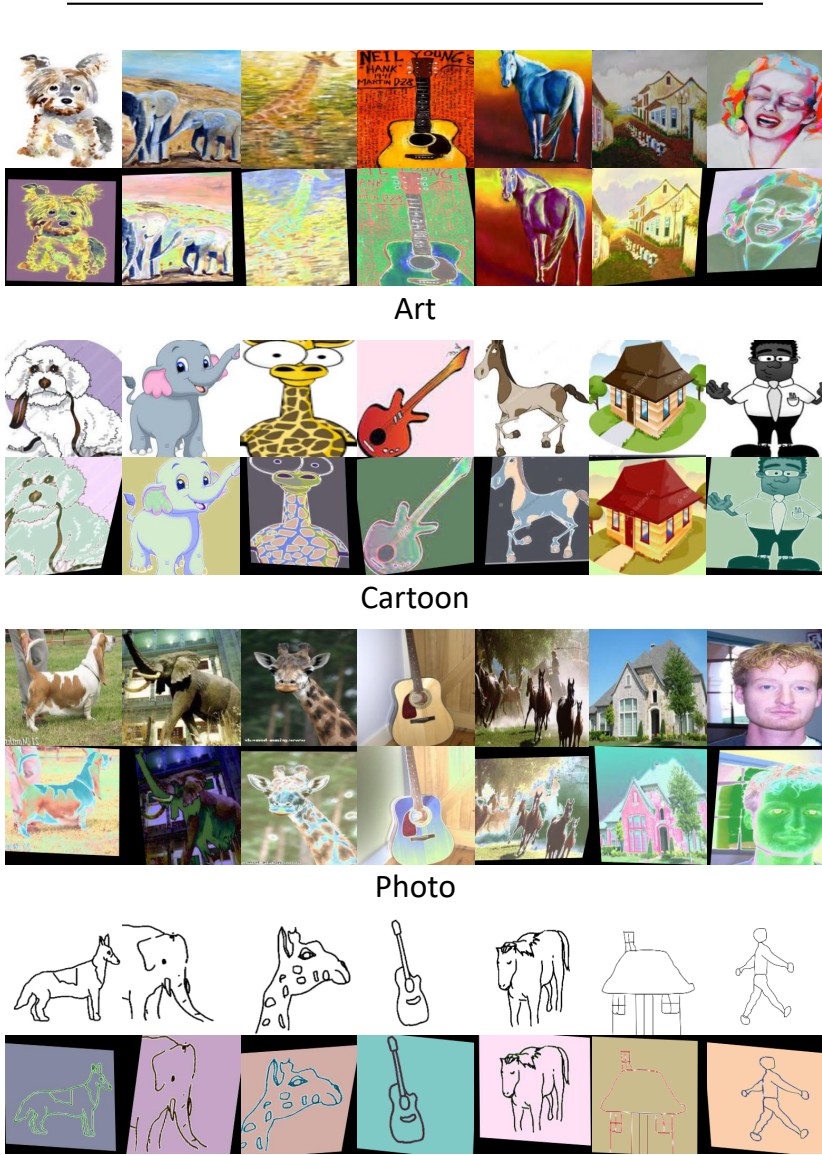

Figure 6: Examples of augmented images on the PACS dataset.

modules, which do not drastically alter the semantic content of images. As illustrated in Fig. 6, the changes primarily affect the "style" or "domain" of the images rather than their semantic meaning.

## D    OTHER DETAILS OF DATASETS AND IMPLEMENTATION

Specifically, Digits-DG is for digits recognition consisting of 4 different digits datasets including MNIST (LeCun et al., 1998), MNIST-M (Ganin & Lempitsky, 2015), SVHN (Netzer et al., 2011), and SYN (Ganin & Lempitsky, 2015), which vary in fonting styles, backgrounds, color, image quality, and so on. For example, SVHN is collected in streets while images of SYN are synthesized.

We follow the train-validation split as in (Zhou et al., 2020), where 480 samples per class per dataset are for training and 120 for testing (24000 samples in total). PACS contains 9,991 images from four different domains (photo, art, cartoon, and sketch) and has 7 object categories mainly about animals. OfficeHome consists of 15,588 images from four different domains (art, clipart, product, and Real-Word) and 56 object categories of everyday objects. Compared to PACS, it has fewer samples per class. The TerraInc dataset has 24,788 images collected from 4 different cameras and 10 object categories of wild animals. Different from PACS and OfficeHome, the objects in images of TerraInc are not always centered. We follow the same train-validation split as in (Zhang et al., 2023b) for PACS, OfficeHome, and TerraInc.

All networks are trained for 40 rounds with 5 local epochs per round, ensuring both local and global convergence as in (Zhang et al., 2023b). We use the SGD optimizer with a batch size of 128 for Digits-DG and 16 for the other datasets. Weight decay is set to 5e-4 for all models. The learning rates are set to 5e-3, 1e-3, and 5e-4 for CNN, ResNet18, and ResNet50, respectively, with decay by a factor of 0.1 at round 32 (i.e., $40 \times 0.8$). For optimization, we use SGD with a batch size of 128 for Digits-DG and 32 for the others. The learning rates for CNN, ResNet18, and ResNet50 are set to 5e-3, 1e-3, and 1e-3, respectively, without decay. For the compared methods, we tune $\mu = 0.1, 1$ for MOON (Li et al., 2021) , $\lambda = 0.1, 0.01, 0.001$ for FedDYN (Acar et al., 2021), and $\lambda = 0.01, 0.02, 0.05, 0.1, \gamma = 0.01, 0.02, 0.05, 0.1, 0.2, 0.5$ for FedSMOO (Sun et al., 2023). For Scaffold (Karimireddy et al., 2020) and FedGAMMA (Dai et al., 2023), we follow the implementation of GA Zhang et al. (2023b), while for other methods, we use hyperparameters as reported in respective papers. In all experiments, we report the mean ($\pm$ std) results based on 3 random runs.

For the augmentation model, the geometry augmentation scale and color augmentation scale are set as 0.125 and 0.2 for Digits-DG, PACS, and OfficeHome and set as 0.0625 and 0.1 for TerraInc. The update interval of the augmentation model is set as 10. The number of data splits is set as 4. The sampling augmentation frequency is set as 10. Other hyperparameters of training augmentation models are kept the same as (Suzuki, 2022). In all experiments, we use an RTX 3090 GPU for training.

### D.1 DETAILS OF FEDIIR

It is important to note that FedIIR and FedIIR (GFM) are not directly comparable to other baselines. FedIIR requires more communication rounds and fewer local epochs to improve gradient estimation and alignment. Specifically, for both FedIIR and GFM (FedIIR), models are trained for 100 rounds with only 1 local epoch per round.

### D.2 AUGMENTATION MODEL

The augmentation model $a(; \phi)$ is defined as the composition of a color augmentation model $c(; \phi_c)$ and a geometry augmentation model $g(; \phi_g)$. The color augmentation model $c(; \phi_c)$ takes $x_i \in \mathbb{R}^{3 \times H \times W}$ as input and outputs color transformation parameters $(\alpha_i, \beta_i)$, where $\alpha_i, \beta_i \in \mathbb{R}^{3 \times H \times W}$ represent scaling and shifting factors, respectively. The augmented color is then computed as $\tilde{x}_i = t(\alpha_i \odot x_i + \beta_i)$, where $t(\cdot)$ denotes a triangle wave function. The geometry augmentation model $g(; \phi_g) : \mathcal{X} \to \mathbb{R}^{2 \times 3}$ also takes $x_i$ as input and outputs a residual affine parameter $A \in \mathbb{R}^{2 \times 3}$. An affine transformation is applied as $\hat{x} = \text{Affine}(\tilde{x}, A + I)$, where $I$ is the identity matrix. The entire procedure is differentiable, and both $\phi_c$ and $\phi_g$ are parameters of deep models. For further details, refer to Section 4 of (Suzuki, 2022).

## E PROOF OF THEOREMS

The proofs of Lemma 1, Lemma 3 and Theorem 1 are done similarly as in (Cha et al., 2021).

### E.1 TECHNICAL LEMMAS

Consider a bounded instance loss function $\ell : \mathcal{Y} \times \mathcal{Y} \to [0, 1]$ such that $\ell(y_1, y_2) = 0$ holds if and only if $y_1 = y_2$. Then we can define the functional error $\mathcal{E}_P(h_1, h_2) := \mathbb{E}_P(\ell(h_1(x), h_2(x)))$. Given two distributions $P$ and $Q$, we have the following lemma.

**Lemma 1.** *The difference between the error with $P$ and the error with $Q$ is bounded by the divergence between $P$ and $Q$:*

$$|\mathcal{E}_P(\ell(h_1, h_2)) - \mathcal{E}_Q(\ell(h_1, h_2))| \leq \frac{1}{2}\mathbf{Div}(P, Q) \tag{A.3}$$

*Proof.* From the Fubini's theorem, we have,

$$\mathbb{E}_{x \in P}[\ell(h_1(x), h_2(x))] = \int_0^\infty \mathbb{P}_P(\ell(h_1(x), h_2(x)) > t)dt \tag{A.4}$$

By using it, we have,

$$|\mathcal{E}_P(\ell(h_1, h_2)) - \mathcal{E}_Q(\ell(h_1, h_2))| \tag{A.5}$$

$$= \left| \int_0^\infty \mathbb{P}_P(\ell(h_1(x), h_2(x)) > t)dt - \int_0^\infty \mathbb{P}_Q(\ell(h_1(x), h_2(x)) > t)dt \right| \tag{A.6}$$

$$\leq \int_0^\infty |\mathbb{P}_P(\ell(h_1(x), h_2(x)) > t) - \mathbb{P}_Q(\ell(h_1(x), h_2(x)) > t)|dt \tag{A.7}$$

$$\leq \sup_{t \in [0,1]} |\mathbb{P}_P(\ell(h_1(x), h_2(x)) > t) - \mathbb{P}_Q(\ell(h_1(x), h_2(x)) > t)| \tag{A.8}$$

$$\leq \sup_{h_1, h_2} \sup_{t \in [0,1]} |\mathbb{P}_P(\ell(h_1(x), h_2(x)) > t) - \mathbb{P}_Q(\ell(h_1(x), h_2(x)) > t)| \tag{A.9}$$

$$\leq \sup_{\overline{h} \in \overline{H}} |\mathbb{P}_P(\overline{h}(x) = 1) - \mathbb{P}_Q(\overline{h}(x) = 1)| \tag{A.10}$$

$$\leq \sup_A |\mathbb{P}_P(A) - \mathbb{P}_Q(A)| = \frac{1}{2}\mathbf{Div}(P, Q) \tag{A.11}$$

where $\overline{H} := \{\mathbb{I}[\ell(h_1(x), h_2(x)) > t]|h_1, h_2; t \in [0,1]\}$. □

**Lemma 2.** *Denote $D := \frac{1}{M}\sum_{i=1}^M D_i$ as the mixture distribution of $M$ source distributions and target distribution $T$, we have:*

$$\mathbf{Div}(D, T) \leq \frac{1}{M}\sum_{i=1}^M \mathbf{Div}(D_i, T). \tag{A.12}$$

*Proof.* From the definition of $\mathbf{Div}(\cdot, \cdot)$, we get,

$$\mathbf{Div}(D, T) = 2\sup_A |\mathbb{P}_D(A) - \mathbb{P}_T(A)| \tag{A.13}$$

$$= 2\sup_A |\frac{1}{M}\sum_{i=1}^M \mathbb{P}_{D_i}(A) - \mathbb{P}_T(A)| \tag{A.14}$$

$$\leq 2\sup_A \frac{1}{M}\sum_{i=1}^M |\mathbb{P}_{D_i}(A) - \mathbb{P}_T(A)| \tag{A.15}$$

$$\leq \frac{1}{M}\sum_{i=1}^M 2\sup_A |\mathbb{P}_{D_i}(A) - \mathbb{P}_T(A)| \tag{A.16}$$

$$= \sum_{i=1}^M \mathbf{Div}(D_i, T) \tag{A.17}$$

□

**Lemma 3.** *Consider a distribution $P$ on input space and global label function $f(\cdot; \theta) : \mathcal{X} \to \mathcal{Y}$. Let $\{\Theta_k \subset \mathbb{R}^d, k = 1, \cdots, N\}$ be a finite cover of a parameter space $\Theta$ which consists of closed balls with radius $\gamma/2$ where $N := \lceil(diam(\Theta)/\gamma)^d\rceil$. Let $\theta_k \in \mathrm{argmax}_{\Theta_k \cap \Theta} \mathcal{E}_P(\theta)$ be a local maximum*

*in the k-th ball. Let a VC dimension of $\Theta_k$ be $v_k$. Then, for any $\theta \in \Theta$, the following bound holds with probability at least $1 - \delta$.*

$$\mathcal{E}_P(\theta) - \hat{\mathcal{E}}_P^\gamma(\theta) \leq \max_k \sqrt{\frac{v_k \left[\ln (n/v_k) + 1\right] + \ln (N/\delta)}{2n}} \tag{A.18}$$

*where $\hat{\mathcal{E}}_P^\gamma(\theta)$ is an empirical robust risk with $n$ samples.*

*Proof.* We first show for the local maximum of N covers, the following inequality holds:

$$\mathbb{P}\left(\max_k \left[\mathcal{E}_P(\theta_k) - \hat{\mathcal{E}}_P(\theta_k)\right] > \epsilon\right) \leq \sum_{k=1}^N \mathbb{P}\left(\mathcal{E}_P(\theta_k) - \hat{\mathcal{E}}_P(\theta_k) > \epsilon\right) \tag{A.19}$$

$$\leq \sum_{k=1}^N \mathbb{P}\left(\sup_{\theta \in \Theta_k} \left[\mathcal{E}_P(\theta) - \hat{\mathcal{E}}_P(\theta)\right] > \epsilon\right) \tag{A.20}$$

$$\leq \sum_{k=1}^N \left(\frac{en}{v_k}\right)^{v_k} e^{-2n\epsilon^2} \tag{A.21}$$

By introducing a confidence error bound $\epsilon_k := \sqrt{\frac{v_k[\ln(n/v_k)+1]+\ln(N/\delta)}{2n}}$ and setting $\epsilon := \max_k \epsilon_k$, we get,

$$\mathbb{P}\left(\max_k \left[\mathcal{E}_P(\theta_k) - \hat{\mathcal{E}}_P(\theta_k)\right] > \epsilon\right) \leq \sum_{k=1}^N \left(\frac{en}{v_k}\right)^{v_k} e^{-2n\epsilon^2} \tag{A.22}$$

$$\leq \sum_{k=1}^N \left(\frac{en}{v_k}\right)^{v_k} e^{-2n\epsilon_k^2} \tag{A.23}$$

$$= \sum_{k=1}^N \frac{\delta}{N} = \delta \tag{A.24}$$

Thus, the $\max_k \left[\mathcal{E}_P(\theta_k) - \hat{\mathcal{E}}_P(\theta_k)\right] \leq \epsilon$ holds with probability at least $1 - \delta$. Based on this, we consider $\mathcal{E}_P(\theta) - \hat{\mathcal{E}}_P^\gamma(\theta)$. For any $\theta$, there exists $k'$ such that $\theta \in \Theta_{k'}$. Then, we get,

$$\mathcal{E}_P(\theta) - \hat{\mathcal{E}}_P^\gamma(\theta) \leq \mathcal{E}_P(\theta) - \hat{\mathcal{E}}_P(\theta_{k'}) \tag{A.25}$$

$$\leq \mathcal{E}_P(\theta) - \mathcal{E}_P(\theta_{k'}) + \epsilon \tag{A.26}$$

$$\leq \mathcal{E}_P(\theta_{k'}) - \mathcal{E}_P(\theta_{k'}) + \epsilon = \epsilon \tag{A.27}$$

Thus, $\mathcal{E}_P(\theta) - \hat{\mathcal{E}}_P^\gamma(\theta) \leq \epsilon$ holds with probability at least $1 - \delta$. $\qquad\square$

**Lemma 4.** *Denote $D := \sum_{i=1}^M p_i D_i$ as the global distribution. The robust risk of the global model $\theta$ is bound by the weighted averaged robust risk of local models, where the weights are combination coefficients:*

$$\hat{\mathcal{E}}_D^\gamma(\theta) \leq \sum_i p_i \hat{\mathcal{E}}_D^\gamma(\theta_i) \tag{A.28}$$

*Proof.* From the Assumption 1 and the definition of $\hat{\mathcal{E}}^\gamma$, we have,

$$\hat{\mathcal{E}}_D^\gamma(\theta) = \hat{\mathcal{E}}_D(\theta + \Delta) \leq \sum_i p_i \hat{\mathcal{E}}_D(\theta_i + \Delta) \leq \sum_i p_i \hat{\mathcal{E}}_D(\theta_i + \Delta_i) = \sum_i p_i \hat{\mathcal{E}}_D^\gamma(\theta_i) \tag{A.29}$$

where $\Delta := \operatorname{argmax}_\Delta \hat{\mathcal{E}}_D(\theta + \Delta)$ and $\Delta_i := \operatorname{argmax}_\Delta \hat{\mathcal{E}}_D(\theta_i + \Delta)$. $\qquad\square$

## E.2 Proof of Theorem 1

**Theorem 1.** *Consider a set of $K$ covers $\{\Theta_k\}_{k=1}^{K}$ such that the parameter space $\Theta \subset \cup_k^K \Theta_k$ where $diam(\Theta) := \sup_{\theta,\theta' \in \Theta} \|\theta - \theta'\|_2$, $K := \left\lceil (diam(\Theta)/\gamma)^d \right\rceil$ and $d$ is dimension of $\Theta$. Let $v_k$ be a VC dimension of each $\Theta_k$. Then, for any $\theta \in \Theta$, the following bound holds with probability at least $1 - \delta$,*

$$\mathcal{E}_T(\theta) < \hat{\mathcal{E}}_{\mathcal{D}^s}^\gamma(\theta) + \frac{1}{2M_s}\sum_{i=1}^{M_s}\mathbf{Div}(D_i, T) + \max_{k \in [1,K]}\sqrt{\frac{v_k \ln(n/v_k) + \ln(K/\delta)}{n}}, \qquad \text{(A.30)}$$

*where $\mathcal{D}^s$ is the set of train (seen) domains, $n$ is the number of training samples per domain, and $\mathbf{Div}(D_i, T) := 2\sup_A |\mathbb{P}_{D_i}(A) - \mathbb{P}_T(A)|$ is a divergence between two distributions.*

*Proof.* Defining the mixture distribution as $D^s := \sum_{i=1}^{M_s} D_i$, we have $\hat{\mathcal{E}}_{\mathcal{D}^s}^\gamma(\theta) = \hat{\mathcal{E}}_{D^s}^\gamma(\theta)$. By applying Lemma 1 (taking $f(\cdot; \theta)$ as $h_1$ and true labeling function as $h_2$), Lemma 3, and Lemma 2 respectively, we get,

$$\mathcal{E}_T(\theta) \leq \mathcal{E}_{D^s}(\theta) + \frac{1}{2}\mathbf{Div}(T, D^s) \qquad \text{(A.31)}$$

$$\leq \hat{\mathcal{E}}_{\mathcal{D}^s}^\gamma(\theta) + \max_{k \in [1,K]}\sqrt{\frac{v_k \ln(n/v_k) + \ln(K/\delta)}{n}} + \frac{1}{2}\mathbf{Div}(T, D^s) \qquad \text{(A.32)}$$

$$\leq \hat{\mathcal{E}}_{\mathcal{D}^s}^\gamma(\theta) + \max_{k \in [1,K]}\sqrt{\frac{v_k \ln(n/v_k) + \ln(K/\delta)}{n}} + \frac{1}{2M_s}\sum_{i=1}^{M_s}\mathbf{Div}(T, D_i) \qquad \text{(A.33)}$$

$$= \hat{\mathcal{E}}_{\mathcal{D}^s}^\gamma(\theta) + \frac{1}{2M_s}\sum_{i=1}^{M_s}\mathbf{Div}(D_i, T) + \max_{k \in [1,K]}\sqrt{\frac{v_k \ln(n/v_k) + \ln(K/\delta)}{n}} \qquad \text{(A.34)}$$

$\square$

## E.3 Proof of Theorem 2

**Theorem 2.** *Denote the local models as $\{\theta_i\}_{i=1}^{M_S}$, the global model as $\theta$, and the augmentation models as $\{\phi_i\}_{i=1}^{M_S}$. Suppose $\{\theta_i\}_{i=1}^{M_S}$ satisfies Assumption 1, $\theta$ is the aggregate of $\{\theta_i\}_{i=1}^{M_S}$ and $p_i = 1/M_s$. For any $\theta \in \Theta$, the following bound holds with probability at least $1 - \delta$:*

$$\mathcal{E}_T(\theta) < \sum_i^{M_s}\frac{1}{M_s}\hat{\mathcal{E}}_{a(D_i;\phi_i)}^\gamma(\theta_i) + \frac{1}{2M_s}\sum_{i=1}^{M_s}\mathbf{Div}(D_i, T) + \max_{k \in [1,K]}\sqrt{\frac{v_k \ln(n/v_k) + \ln(K/\delta)}{n}},$$
$$\text{(A.35)}$$

*where $\phi_i = \underset{\phi_i}{\arg\max}\, \hat{\mathcal{E}}_{a(D_i;\phi_i)}(\theta_i + \Delta_i) - \hat{\mathcal{E}}_{a(D_i;\phi_i)}(\theta)$.*

*Proof.* Defining the mixture distribution as $D^s := \sum_{i=1}^{M_s} D_i$, we get,

$$\mathcal{E}_T(\theta) \leq \mathcal{E}_{D^s}(\theta) + \frac{1}{2}\mathbf{Div}(T, D^s) \tag{A.36}$$

$$\leq \hat{\mathcal{E}}_{\mathcal{D}^s}^\gamma(\theta) + \max_{k \in [1,K]} \sqrt{\frac{v_k \ln(n/v_k) + \ln(K/\delta)}{n}} + \frac{1}{2}\mathbf{Div}(T, D^s) \tag{A.37}$$

$$\leq \frac{1}{M_s} \sum_i^{M_s} \hat{\mathcal{E}}_{\mathcal{D}^s}^\gamma(\theta_i) + \max_{k \in [1,K]} \sqrt{\frac{v_k \ln(n/v_k) + \ln(K/\delta)}{n}} + \frac{1}{2}\mathbf{Div}(T, D^s) \tag{A.38}$$

$$\leq \frac{1}{M_s} \sum_i^{M_s} \hat{\mathcal{E}}_{a(D_i;\phi_i)}(\theta_i + \Delta_i) + \max_{k \in [1,K]} \sqrt{\frac{v_k \ln(n/v_k) + \ln(K/\delta)}{n}} + \frac{1}{2}\mathbf{Div}(T, D^s) \tag{A.39}$$

$$\leq \frac{1}{M_s} \sum_i^{M_s} \hat{\mathcal{E}}_{a(D_i;\phi_i)}^\gamma(\theta_i) + \max_{k \in [1,K]} \sqrt{\frac{v_k \ln(n/v_k) + \ln(K/\delta)}{n}} + \frac{1}{2}\mathbf{Div}(T, D^s) \tag{A.40}$$

$$\frac{1}{M_s} \sum_i^{M_s} \hat{\mathcal{E}}_{a(D_i;\phi_i)}^\gamma(\theta_i) + \frac{1}{2M_s} \sum_{i=1}^{M_s} \mathbf{Div}(D_i, T) + \max_{k \in [1,K]} \sqrt{\frac{v_k \ln(n/v_k) + \ln(K/\delta)}{n}} \tag{A.41}$$

where $\Delta_i := \mathrm{argmax}_\Delta \, \hat{\mathcal{E}}_D(\theta_i + \Delta)$. The second inequality holds since Lemma 4 and the third inequality holds because of Eq. (12). □

## F ANALYSIS OF MIN-MAX OPTIMIZATION

By replacing robust risk and empirical risk with population risk, we fucus on a simplified min-max optimization objective, denoted as $\Delta\mathcal{E}(\theta_i, \phi_i)$, as follows:

$$\min_{\theta_i} \max_{\phi_i} \left[ \mathcal{E}_{a(D_i;\phi_i)}(\theta_i) - \mathcal{E}_{a(D_i;\phi_i)}(\theta) \right]. \tag{A.42}$$

Let $q(x; \phi_i)$ denote the probability density function of $a(D_i; \phi_i)$, $p(y|x; \theta)$ represent the prediction probability of the true label $y$ given $\theta$, $p(y|x; \theta_i)$ represent the prediction probability given $\theta_i$, and $\ell(\cdot)$ denote the cross-entropy loss. The equation can then be reformulated as:

$$\min_{\theta_i} \max_{\phi_i} \int q(x; \phi_i)\big( \ln p(y|x; \theta) - \ln p(y|x; \theta_i)\big)dx. \tag{A.43}$$

To begin, we consider the optimal solution of the maximization process. Define $X^* = \{x \mid x = \mathrm{argmax}_x \big( \ln p(y|x; \theta) - \ln p(y|x; \theta_i)\big)\} = \{x_j^*\}$. Assuming the augmentation model is a sufficiently powerful model with enough capacity, the optimal solution satisfies:

$$q(x; \phi_i) = \sum_{j=1}^{|X^*|} w_j \delta(x - x_j^*), \tag{A.44}$$

where $\delta$ denotes the Dirac function, $w_j \geq 0$, and $\sum_j w_j = 1$. Intuitively, the augmentation model aims to generate samples that exhibit the largest prediction discrepancy with respect to the true label. In the subsequent minimization step, $\theta_i$ seeks to improve its performance on these challenging samples. Through this process, $\theta_i$ is guided to align its behavior with $\theta$. The process continues until $\theta_i$ can no longer improve its performance on these samples, and/or until the models reach a certain equilibrium. Proposition 1 presents a saddle point solution for the min-max process.

**Proposition 1.** *A saddle point solution exists for the min-max problem in Equation* (A.42)*. Construct $\theta_i^*$ such that $p(y|x; \theta_i^*) = s \cdot p(y|x; \theta)$ for any $x$ in the support set with true label $y$, where $s = \frac{1}{\max_x p(y|x;\theta)}$. Then, there exists $\phi_i^*$ such that $\theta_i^*$ is the local minimum of $\mathcal{E}_{a(D_i;\phi_i^*)}(\theta_i^*)$. Consequently, the pair $(\theta_i^*, \phi_i^*)$ constitutes a saddle point solution, satisfying $\Delta\mathcal{E}(\theta_i^*, \phi_i) \leq \Delta\mathcal{E}(\theta_i^*, \phi_i^*) \leq \Delta\mathcal{E}(\theta_i, \phi_i^*)$.*

*Proof.* We begin with the construction of $\phi_i^*$. Denoting $\bar{X} = \{x | x = \text{argmax}_x \, p(y|x; \theta)\} = \{\bar{x}_j\}$, we construct $\phi_i^*$ that satisfies:

$$q(x; \phi_i^*) = \sum_{j=1}^{|\bar{X}|} w_j \delta(x - \bar{x}_j), \tag{A.45}$$

where $\delta$ indicates the Dirac function, $w_j \geq 0$ and $\sum_j w_j = 1$. As a result,

$$\mathcal{E}_{a(D_i; \phi_i^*)}(\theta_i^*) = \int q(x; \phi_i^*) \ln p(y|x; \theta_i^*) dx \tag{A.46}$$

$$= \sum_{j=1}^{|\bar{X}|} w_j \ln s \cdot p(y|\bar{x}; \theta) \tag{A.47}$$

$$= 0. \tag{A.48}$$

Because cross-entropy is a convex function of prediction and $\mathcal{E}_{a(D_i; \phi_i^*)}(\theta_i^*)$ reaches the optimal value, $\theta_i^*$ is a minimum. Thus, $\Delta\mathcal{E}(\theta_i^*, \phi_i^*) \leq \Delta\mathcal{E}(\theta_i, \phi_i^*)$ holds. Then, to prove $\Delta\mathcal{E}(\theta_i^*, \phi_i) \leq \Delta\mathcal{E}(\theta_i^*, \phi_i^*)$, we just need to prove that $\bar{x} \in X^*$ according to Equation (A.44). Because $\ln p(y|x; \theta) - \ln p(y|x; \theta_i^*) = \ln s$ for any $x$ in the support set, $\bar{x} \in X^*$ is valid. So, we get $\Delta\mathcal{E}(\theta_i^*, \phi_i) \leq \Delta\mathcal{E}(\theta_i^*, \phi_i^*)$, which concludes the proof. $\square$

The min-max process converges to the saddle point once it reaches its neighborhood. The saddle point solution in Proposition 1 demonstrates comparable global performance to the global model $\theta$, as shown by: $p(y|x; \theta_i^*) = s \cdot p(y|x; \theta)$. In this way, the local update can be effectively supplemented with global information, leveraging both the global model and the augmentation model.

Notably, a low value of $s$ may limit the performance improvement of $\theta_i^*$. To address this limitation, an auxiliary conditional distribution can be defined as:

$$p_a(y|x) = \min(t, p(y|x; \theta)), \tag{A.49}$$

where $t \in (0, 1)$ is a hyperparameter. Using this auxiliary distribution, the final modified min-max problem becomes:

$$\min_{\theta_i} \max_{\phi_i} \int q(x; \phi_i)\big(\ln p_a(y|x) - \ln p(y|x; \theta_i)\big) dx. \tag{A.50}$$