# OpenReview forum: "Seeking Global Flat Minima in Federated Domain Generalization via Constrained Adversarial Augmentation"
_ICLR.cc/2025/Conference — Submitted to ICLR 2025_

### Official Review · Reviewer_iTLS · 2024-10-30

**Soundness:** 3
**Presentation:** 4
**Contribution:** 3
**Rating:** 8
**Confidence:** 5

**Summary:**

The paper addresses the challenge of federated domain generalization (FedDG), where a federated model must generalize effectively to new clients facing domain shifts. Authors introduce GFM, an algorithm designed to achieve global flat minima during the federated training of global model. GFM uses a globally-constrained adversarial data augmentation technique, which generates surrogate global data within each local client, improving the model’s flatness while preserving data privacy. Supported by theoretical analysis, this approach ensures that GFM effectively finds global flat minima. Extensive experiments further highlight GFM’s superior performance over previous FL and FedDG methods across multiple benchmarks.

**Strengths:**

1.	The exploration of the global loss landscape within FedDG research is novel, and the proposed approach of enhancing global flatness through data augmentation represents an innovative and promising direction.
2.	Theoretical analysis in this paper is rigorous, skillfully incorporating global flatness into local updates. This theoretical foundation effectively guides the algorithm design, resulting in a well-grounded and interpretable solution.
3.	Claims made in the methodology, including Assumption 1 and those related to model flatness, are well-validated and substantiated by experimental results.

**Weaknesses:**

1.	The min-max optimization process in the local updates remains somewhat unclear. Further clarification would enhance understanding of the authors’ claim.
2.	This paper lacks direct validation to demonstrate whether the augmented data provides a more effective surrogate for global data compared to local data. A more focused assessment on this point would strengthen the findings.
3.	This paper does not include an analysis of hyperparameter selection, which is important for SAM-based approaches.

**Questions:**

1.	Could the authors provide a more detailed analysis of the min-max process? For instance, analyzing the solutions of $\phi_i$ and $\theta_i$ would improve understanding of this component.
2.	Regarding point 2 in Weaknesses: Does the augmented data act as a better surrogate for global data than local data? A more direct evaluation would be helpful.
3.	A sensitivity analysis of key hyperparameters, such as the radius $\gamma$, would add valuable insight.
4.	The work by Sun et al. [1] also explores global flatness in heterogeneous FL using an ADMM algorithm to address constrained optimization. Including a citation of this work and a comparison would enhance the paper.

[1] Sun, Yan, et al. “Dynamic Regularized Sharpness Aware Minimization in Federated Learning: Approaching Global Consistency and Smooth Landscape.” International Conference on Machine Learning. PMLR, 2023.

---

> ### Author Response · Authors · 2024-11-22
> **Response to Reviewer iTLS [1/2]**
>
> ## W1 & Q1
>
> > W1: The min-max optimization process in the local updates remains somewhat unclear. Further clarification would enhance understanding of the authors’ claim.
> > Q1: Could the authors provide a more detailed analysis of the min-max process? For instance, analyzing the solutions of $\phi_i$ and $\theta_i$ would improve understanding of this component.
>
> Thanks for your valuable advice. We provide a more detailed analysis of the min-max process in the new revision as shown in Appendix F, including:
>
> - **The optimal solution of $\phi_i$:**  we show that the augmentation model aims to generate samples that exhibit the largest prediction discrepancy with respect to the true label. In the subsequent minimization step, $\theta_i$ seeks to improve its performance on these challenging samples. Through this process, $\theta_i$ is guided to align its behavior with $\theta$.
> - **Existence of the desirable saddle point:**  we prove the existence of the saddle point in the min-max process in Proposition 1. It can be inferred that the saddle point solution in Proposition 1 is desirable because it achieves comparable global performance to the global model $\theta$, as demonstrated by $p(y|x;\theta_i^*) = s \cdot p(y|x;\theta)$. In this way, the local update can be effectively supplemented with global information as stated, leveraging both the global model and the augmentation model.
>
> > Proposition 1: A saddle point solution exists for the min-max problem in Eq. (14). Construct $\theta_i^*$ such that $p(y|x;\theta_i^*) = s \cdot p(y|x;\theta)$ for any $x$ in the support set with true label $y$, where $s = \frac{1}{\max_x p(y|x;\theta)}$. Then, there exists $\phi_i^*$ such that $\theta_i^*$ is the local minimum of $\mathcal{E}_{a(D_i;\phi_i^*)}(\theta_i^*)$. Consequently, the pair $(\theta_i^*, \phi_i^*)$ constitutes a saddle point solution, satisfying $ \Delta \mathcal{E}(\theta_i^*,\phi_i)\leq\Delta \mathcal{E}(\theta_i^*,\phi_i^*)\leq \Delta \mathcal{E}(\theta_i,\phi_i^*)$.
>
>
> ## W2 & Q2
>
> > W2: This paper lacks direct validation to demonstrate whether the augmented data provides a more effective surrogate for global data compared to local data. A more focused assessment on this point would strengthen the findings.
> >
> > Q2: Regarding point 2 in Weaknesses: Does the augmented data act as a better surrogate for global data than local data? A more direct evaluation would be helpful.
>
>
> To provide a more direct evaluation, we focus on the trained model after local updates. As shown in Proposition 1, the trained model should perform similar to the global model. Consequently, we evaluate the effect of augmented data and local data by measuring the mean forgetting rate between the trained model and the global model on the global dataset $\hat{D}$ as shown in Appendix C.2. There are two major conclusions:
>
> - It is evident that **GCA adopted in GFM effectively ameliorates the situation of forgetting**.
> - The amelioration is more evident in the later stages of training **when the global model becomes stronger**.

---

> ### Author Response · Authors · 2024-11-22
> **Response to Reviewer iTLS [2/2]**
>
> ## W3 & Q3
>
> > W3: This paper does not include an analysis of hyperparameter selection, which is important for SAM-based approaches.
> >
> > Q3: A sensitivity analysis of key hyperparameters, such as the radius γ, would add valuable insight.
>
> Thanks for your valuable advice. In the new revision, we provide Parameter analysis in Section 4.6.
>
> | $\gamma$ | 0.002 | 0.005 | 0.01  | 0.02  | 0.05  |  0.1  |
> | -------- | :---: | :---: | :---: | :---: | :---: | :---: |
> | FedSAM   | 82.62 | 82.90 | 83.14 | 83.36 | 82.70 | 82.72 |
> | GFM      | 83.84 | 84.20 | 84.44 | 84.46 | 84.05 | 83.23 |
>
> Major points related to the selection of $\gamma$ is concluded as follows:
>
> - **The selection of $\gamma$ is based on the grid search:**  the optimal value of $\gamma$ varies across tasks, datasets, and models for SAM-based methods. As a result, we conducted a grid search for $\gamma$ on the PACS dataset to determine the appropriate value for both FedSAM and GFM. The local optimum ($\gamma=0.02$) exists for both methods.
> - **GFM exhibits a relatively flatter optimum compared to FedSAM:**  this could be attributed to the improved consistency of local models in GFM, which reduces the need for local flatness to achieve sufficient global flatness.
>
> ## Q4
>
> > Q4: The work by Sun et al. [1] also explores global flatness in heterogeneous FL using an ADMM algorithm to address constrained optimization. Including a citation of this work and a comparison would enhance the paper.
>
> In this revision, we cited [1] in the Introduction Section and compared to it in Table 1.
>
> | Method        |     Digits-DG      |        PACS        |     OfficeHome     |      TerraInc      |   Avg.    |
> | :------------ | :----------------: | :----------------: | :----------------: | :----------------: | :-------: |
> |               |      ConvNet       |      ResNet18      |      ResNet18      |      ResNet50      |           |
> | FL methods    |                    |                    |                    |                    |           |
> | FedAvg        |   67.46$\pm$0.27   |   82.56$\pm$0.47   |   64.82$\pm$0.28   |   44.23$\pm$0.69   |   64.77   |
> | Scaffold      |   68.30$\pm$0.79   |   82.54$\pm$0.25   |   64.56$\pm$0.20   |   42.70$\pm$0.46   |   64.53   |
> | FedDyn        |   68.18$\pm$0.14   |   82.73$\pm$0.24   |   63.89$\pm$0.13   |   44.28$\pm$0.71   |   64.77   |
> | MOON          |   65.79$\pm$0.98   |   82.65$\pm$0.53   |   62.87$\pm$0.13   |   43.73$\pm$0.77   |   63.76   |
> | FedSAM        |   66.67$\pm$0.49   |   83.36$\pm$0.22   |   65.28$\pm$0.35   |   45.16$\pm$1.36   |   65.12   |
> | FedGAMMA      |   67.70$\pm$1.54   |   82.83$\pm$0.34   |   65.38$\pm$0.12   |   43.56$\pm$1.04   |   64.87   |
> | FedSMOO       |   69.43$\pm$0.53   |   82.92$\pm$0.79   |   62.40$\pm$0.22   |   43.38$\pm$0.76   |   64.53   |
> | FedDG methods |                    |                    |                    |                    |           |
> | FedSR         |   68.21$\pm$0.38   |   83.20$\pm$0.83   |   63.99$\pm$0.31   |   42.97$\pm$0.93   |   64.59   |
> | GA            |   68.45$\pm$0.16   |   83.39$\pm$0.61   |   65.11$\pm$0.05   |   45.59$\pm$0.98   |   65.64   |
> | StableFDG     |   67.80$\pm$0.89   |   84.22$\pm$0.72   |   64.61$\pm$0.02   |   44.48$\pm$0.14   |   65.28   |
> | FedIIR        |   69.25$\pm$0.25   |   83.94$\pm$0.16   |   60.64$\pm$0.33   |   46.88$\pm$0.80   |   65.18   |
> | GFM           |  *69.72$\pm$0.99*  |   84.46$\pm$0.42   |  *65.57$\pm$0.19*  |   46.02$\pm$1.04   |  *66.44*  |
> | GFM (GA)      | **71.32$\pm$0.64** | **84.97$\pm$0.22** | **66.08$\pm$0.20** |  *46.91$\pm$0.54*  | **67.32** |
> | GFM (FedIIR)  |   69.57$\pm$1.12   |  *84.67$\pm$0.40*  |   61.74$\pm$0.36   | **47.66$\pm$0.82** |   65.91   |

---

> > ### Comment · Reviewer_iTLS · 2024-11-23
> > **Thanks for the reply.**
> >
> > Thanks for the response. After reading the authors’ thorough rebuttal and other reviewers’ comments, I feel the authors have well addressed my concerns and thus will increased my score.
> >
> > Best, The reviewer

---

> > > ### Author Response · Authors · 2024-11-25
> > >
> > > Thank you very much for your quick response and the positive evaluation. We are pleased to hear that we have addressed all your concerns.

---

### Official Review · Reviewer_hj24 · 2024-11-02

**Soundness:** 3
**Presentation:** 3
**Contribution:** 3
**Rating:** 6
**Confidence:** 3

**Summary:**

The paper proposes a method GFM that flattens the global model to a minimum in the case of federated learning. the paper proposes that the method of finding perspectives on global data is based on a solid theoretical foundation that improves the generalization of the distributional bias of a new client.

**Strengths:**

1. the paper's provides a sound and complete theoretical foundation for their method.
2. the paper provides sufficient experiments to show that their method is better and more suitable than FedSAM to be applied to the federated learning method of divisional offsets.
3. the paper provides sufficient literature, citations to illustrate the relevant issues.

**Weaknesses:**

Assumption 1 of the paper mentions that local models may outperform global models is COUNTER-INTUITIVE, and I think this hypothesis can be illustrative to a certain extent, but it does not cover so the federated learning situation. Because the local model is stronger than the global model in the federated learning chaotic situation is not a completely impossible situation.

**Questions:**

1. It is not particularly clear to me what is the difference between the paper's approach and the client-side optimization of local models using SAM-like methods?

2. As mentioned in the article, such methods require more power on the client side. The computational paper on how to reduce client-side computation doesn't go much further than that

---

> ### Author Response · Authors · 2024-11-22
> **Response to Reviewer hj24**
>
> ## W1
>
> > Assumption 1 of the paper mentions that local models may outperform global models is COUNTER-INTUITIVE, and I think this hypothesis can be illustrative to a certain extent, but it does not cover so the federated learning situation. Because the local model is stronger than the global model in the federated learning chaotic situation is not a completely impossible situation.
>
> This is a very good point. In certain chaotic federated learning scenarios, it is indeed possible for a local model to outperform the global model. For example, this could occur if a local client has a large amount of data with a distribution similar to the global data, while other clients have limited data or significantly differing distributions.
>
> However, we mainly focus on the **federated domain generalization (FedDG) setting**, where each node contains data from **a single domain with sufficient volume**. In this context, it is almost impossible for a model trained on one domain to perform well across all domains.  As such, we believe that Assumption 1 holds under the FedDG setting presented in the paper. For scenarios where the assumption might not hold, caution should be exercised when applying our method, as the global model might impose incorrect constraints in such cases.
>
> We have revised Assumption 1 to include additional constraints, as follows:
>
> > **Assumption 1**. If (1) data distributions $\{D_i\}\_{i=1}^{M_S}$ across clients exhibit a non-trivial degree of heterogeneity, and (2) each client has access to a sufficiently large local dataset to estimate the data distribution. Then during the training phase, local models $\{\theta_i\}\_{i=1}^{M_S}$ and their aggregate $\sum_i p_i \theta_i$​, when weighted by coefficients specific to clients, satisfy the following inequality:
> > $$
> > \hat{\mathcal{E}}\_D(\sum_i p_i \theta_i)\leq\sum_ip_i\hat{\mathcal{E}}\_D(\theta_i),
> > $$
> > where $p_i$ represents the coefficient of client $i$.
>
> ## Q1
>
> > It is not particularly clear to me what is the difference between the paper's approach and the client-side optimization of local models using SAM-like methods?
>
> Thank you for your question. One key conclusion from Theorem 2 is that **the performance on unseen domains is directly tied to the local flatness of seen clients on augmented samples**. This suggests that instead of designing a specific global flatness strategy, we can effectively combine any existing local flatness minimizer with the proposed GCA.
>
> As a result, our algorithm integrates a client-side flatness optimization method with the proposed GCA. The **presence or absence of GCA** is the critical distinction between GFM and purely client-side optimization methods. Moreover, GFM is designed to achieve **global flat minima**, whereas local flatness methods like SAM focus only on finding local flat minima.
>
> ## Q2
>
> > As mentioned in the article, such methods require more power on the client side. The computational paper on how to reduce client-side computation doesn't go much further than that.
>
> In the revised version, we further optimized training speed to reduce computational costs on the client side by implementing the following strategies:
>
> - **Parallel Augmentation:** Previously, the basic augmentation in GFM was implemented serially. In the revised version, we have optimized this by implementing parallel augmentation.
>
> - **Disable Gradients for Classification Model Parameters During Augmentation Updates:** Gradients for the classification model's parameters are now disabled when updating the augmentation model, reducing computational overhead.
>
> As a result of these improvements, the augmentation cost has been reduced from 1.36V to 1.16V (with the total cost reduced from 3.06V to 2.86V), taking the cost of FedSAM (1.7V) as the baseline cost for the classification model. To further reduce the computational cost, we **use subsampled low-dimension images as the inputs to the augmentation model.** The resulting color and affine parameters are then applied to the original images to generate the augmented images, as discussed in Appendix C.5.
>
> | Image Dimension | $28\times 28$ | $56\times 56$ | $112\times 112$ | $224\times 224$ |
> | :-------------: | :-----------: | :-----------: | :-------------: | :-------------: |
> |      Time       |    $2.30V$    |    $2.31V$    |     $2.44V$     |     $2.86V$     |
> |       ACC       |     84.35     |     84.30     |      84.37      |      84.46      |
>
> Fortunately, **the final performance is not significantly impacted by the input image dimensions** used in the augmentation model. By adopting this strategy, the augmentation cost reduces from 1.16V to 0.6 V, which only accounts for 35% of the classification cost. Moreover, as shown in Table 7, among the methods designed to achieve flatter minima for the global model (including FedGAMMA, FedSMOO, and GFM), our method imposes the **lowest space constraints**.

---

> > ### Author Response · Authors · 2024-11-25
> > **Kind Reminder to Review Our Rebuttal**
> >
> > Dear Reviewer hj24,
> >
> > We sincerely appreciate the effort and time you have devoted to providing constructive reviews, as well as your positive evaluation of our submission. As the deadline for discussion and paper revision is approaching,  we would like to offer a brief summary of our responses and updates:
> >
> > - Supplementary constraints for Assumption 1.
> > - Further clarification for the difference between the our approach and the client-side optimization of local models using SAM-like methods.
> > - Details on the acceleration strategies employed in our algorithm and their effects.
> >
> > **Would you mind checking our responses and confirming if you have any additional questions? We welcome any further comments and discussions!**
> >
> > Best Regards,
> >
> > The Authors

---

> > ### Comment · Reviewer_hj24 · 2024-11-25
> > **Thanks for your response**
> >
> > Thanks for the detailed responses. I believe the authors should continue to maintain a cautious attitude when introducing Assumption 1 to ensure it aligns with the specific domain of their research. I will maintain my score.

---

> > > ### Author Response · Authors · 2024-11-25
> > >
> > > Thank you very much for your response and the positive evaluation. We will continue to maintain a cautious attitude when introducing Assumption 1 as you suggest.

---

### Official Review · Reviewer_E99p · 2024-11-10

**Soundness:** 3
**Presentation:** 2
**Contribution:** 2
**Rating:** 5
**Confidence:** 3

**Summary:**

This paper proposes a federated domain generalization method, GFM, which incorporates techniques from sharpness-aware learning. The authors aim to find a flat minimum by coordinating clients during federated training. In addition to sharpness-aware learning, a global model-constrained adversarial data augmentation strategy is employed to better regulate the sharpness of the converged models. The authors claim that existing methods overlook data privacy concerns, motivating the design of the GFM training algorithm. A series of experiments is conducted to verify its effectiveness.

**Strengths:**

1. The paper provides rigorous preliminaries and thorough theoretical support, with detailed proofs included in the appendix.

2. A series of experiments are conducted here to verify the effectiveness of the proposed method.

3. The proposed method is tailored for domain generalization tasks in both its theoretical derivation and algorithm design.

**Weaknesses:**

1. The proposed method is very similar to the baseline, FedSAM; however, it has not been discussed in either the related works section or the introduction. In comparing the proposed method to FedSAM, I do not see a clear improvement or significant difference between them. The only notable difference appears to be the augmentation network discussed around Equation 9, which is directly adopted from Suzuki (2022). Additionally, the ablation study does not evaluate the performance impact of removing this augmentation network.

2. What is the primary motivation of this paper? In the introduction, the authors discuss issues related to finding flat minima and the neglect of data privacy. However, there does not appear to be any specific design or consideration aimed at enhancing privacy protection in this paper.

3. The organization of this paper could be improved. The related works and preliminaries sections span two pages, while the experimental section only covers three pages. The experimental section should be expanded with more detail in the main text.

4. It would be better to add hyperlinks to the citations for the methods listed in all the tables.
Currently, all methods are cited only in Section 5.2, which makes it difficult to link each method to its corresponding paper.

5. The definition of flatness in Equation 3 is based on the assumption that $\theta$ is close to a local minimum, meaning the first-order gradient is near zero. There is no such assumption around equation 3.

**Questions:**

1. How is the loss landscape illustrated in Figure 2? What method is used to generate the loss landscape in this figure?

2. What would the performance be if FedSAM were combined with the augmentation network? Additionally, is there an ablation study on the augmentation network's impact within GFM?

3. Did the authors conduct experiments on CIFAR-10 or CIFAR-100, as some baselines reported results on these datasets?

4. The authors mentioned 'However, in federated learning, this becomes challenging due to the privacy concerns of the data.', which seems to be one of the motivations to derive the proposed GFM. What is the difference of the privacy protection between the GFM and FedSAM?

---

> ### Author Response · Authors · 2024-11-22
> **Response to Reviewer E99p [1/2]**
>
> ## W1 & Q2
>
> > W1.1: The proposed method is very similar to the baseline, FedSAM; however, it has not been discussed in either the related works section or the introduction.
>
> Since there are relatively few studies in the literature focusing on seeking flat minima under federated learning settings, we did not create a separate subsection in the related work section. Instead, we integrated this discussion into the latter part of the flat minima subsection, which might have made it less noticeable to readers. In the revised version, we have clarified this part and FedSAM in the third paragraph of the Introduction section to improve visibility and understanding.
>
> > W1.2: In comparing the proposed method to FedSAM, I do not see a clear improvement or significant difference between them.
>
> Regarding the relationship with FedSAM, we do not view our method as merely a refinement of the specific FedSAM approach. As demonstrated in Theorem 2, the performance on the unseen domain is directly related to the **local flatness of seen clients on augmented samples**. This insight suggests that instead of designing a specific global flatness strategy, we can **combine any existing local flatness optimizer with the proposed GCA**, which we consider a key strength and contribution of our work. We selected SAM primarily due to its simplicity and widespread adoption. It is also worth noting that the SAM optimization scheme was not originally proposed by FedSAM, which explains why we did not specifically emphasize FedSAM in the paper.
>
> We show in the following that our method could combine with another local flatness optimizer. Specifically, we use SWAD [1] to replace SAM in local training ($n_{converge}=5,n_{tolerance}=6$ and the ratio of tolerance as $0.2$).
>
> | Method    |    Art    |  Cartoon  |   Photo   |  Sketch   | Avg.  |
> | --------- | :-------: | :-------: | :-------: | :-------: | :---: |
> | FedAvg    |   77.41   |   77.82   |   92.67   |   82.35   | 82.56 |
> | FedSAM    |   78.34   | **78.85** |   92.45   |   83.81   | 83.36 |
> | GFM(SWAD) | **80.71** |   78.29   | **93.20** |   84.57   | 84.19 |
> | GFM(SAM)  |   80.34   |   78.27   |   92.53   | **86.70** | 84.46 |
>
> To summary, the difference between two methods are as follows:
>
> - While both GFM and FedSAM share the SAM component, **their focuses are fundamentally different**. GFM emphasizes achieving flatness for the global model, whereas FedSAM focuses solely on local flatness.
> - Our proposed augmentation strategy (GCA) can be **flexibly combined with other local flatness optimizers**, such as SWAD, in place of SAM.
>
> > W1.3: The only notable difference appears to be the augmentation network discussed around Equation 9, which is directly adopted from Suzuki (2022).
>
> About the augmentation network, though the network structure was proposed in Suzuki (2022), we only borrowed its network structure (similar to borrowing ResNet for classification tasks). The global model-constrained adversarial data augmentation objective (Eq. (14)), however, is specifically d**esigned for the global flatness objective**. Therefore, we believe this does not overshadow our contribution.
>
> > W1.4: Additionally, the ablation study does not evaluate the performance impact of removing this augmentation network.
> >
> > Q2: What would the performance be if FedSAM were combined with the augmentation network? Additionally, is there an ablation study on the augmentation network's impact within GFM?
>
> When the augmentation network is removed, our method reduces to FedAvg+SAM, which is essentially FedSAM as discussed in line 480. What's more, results of GCA in the ablation study reveal the augmentation network's impact within GFM.
>
> | Method |  Digits   |   PACS    | OfficeHome | TerraInc  |   Avg.    |
> | ------ | :-------: | :-------: | :--------: | :-------: | :-------: |
> | FedAvg |   67.46   |   82.56   |   64.82    |   44.23   |   64.77   |
> | FedSAM |   66.67   |   83.36   |   65.28    |   45.16   |   65.12   |
> | GCA    |   69.65   |   83.91   |   64.64    |   44.20   |   64.93   |
> | GFM    | **69.72** | **84.46** | **65.57**  | **46.02** | **66.44** |
>
> ### Reference
>
> [1] Cha, Junbum, et al. "Swad: Domain generalization by seeking flat minima." *Advances in Neural Information Processing Systems* 34 (2021): 22405-22418.
>
> ## W2
>
> > What is the primary motivation of this paper? In the introduction, the authors discuss issues related to finding flat minima and the neglect of data privacy. However, there does not appear to be any specific design or consideration aimed at enhancing privacy protection in this paper.
>
> The primary motivation is to seek flat minima of the global model to improve domain generalization ability. To avoid violating data protection principles, we adopted the GCA component to generate a surrogate for the desired global data distribution in each local clients.

---

> ### Author Response · Authors · 2024-11-22
> **Response to Reviewer E99p [2/2]**
>
> ## W3
>
> > The organization of this paper could be improved. The related works and preliminaries sections span two pages, while the experimental section only covers three pages. The experimental section should be expanded with more detail in the main text.
>
> Thank you for your kind suggestion. We will revise the related works section by moving content related flatness in federated learning setting to the introduction Section, placing other parts in the Appendix A. In the meanwhile, we add more experiments in the experimental section.
>
> ## W4
>
> > It would be better to add hyperlinks to the citations for the methods listed in all the tables. Currently, all methods are cited only in Section 5.2, which makes it difficult to link each method to its corresponding paper.
>
> Thanks for your suggestion. We have added hyperlinks in Table 1.
>
> ## W5
>
> > The definition of flatness in Equation 3 is based on the assumption that θ is close to a local minimum, meaning the first-order gradient is near zero. There is no such assumption around equation 3.
>
> Thanks for your comment. We apologize for the imprecise expression. $\mathcal{E}^\gamma(\theta)$ is not a direct measurement when $\theta$ is not close to a local minimum but is a promising objective to seek flat minima, as used in [2]. So we adopt  $\mathcal{E}^\gamma(\theta)$ as our objective in the paper. We have modified the corresponding expression in this revision.
>
> [2] Foret, Pierre, et al. "Sharpness-aware minimization for efficiently improving generalization." *arXiv preprint arXiv:2010.01412* (2020).
>
> ## Q1
>
> > How is the loss landscape illustrated in Figure 2? What method is used to generate the loss landscape in this figure?
>
> We use a visualization technique similar to that described in Garipov et al. (2018). Specifically, three local models are used to define the averaging plane, where each point represents a set of model parameters. The value at each point corresponds to the empirical risk of the associated model. By setting the global model and model $a$ as X-axis and model $b$ in the upper of X, the plot is generated.
>
> ## Q3
>
> > Did the authors conduct experiments on CIFAR-10 or CIFAR-100, as some baselines reported results on these datasets?
>
> Because the focus of our paper is **federated domain generalization**, we didn't conduct experiments on CIFAR-10 or CIFAR-100 which only involves a single domain. We introduced these baselines of heterogeneous FL for providing a broader range of references on FedDG problem.
>
> ## Q4
>
> > The authors mentioned 'However, in federated learning, this becomes challenging due to the privacy concerns of the data.', which seems to be one of the motivations to derive the proposed GFM. What is the difference of the privacy protection between the GFM and FedSAM?
>
> There is no difference of privacy protection between GFM and FedSAM and both methods adhere to the principle of data privacy protection. We mentioned 'However, in federated learning, this becomes challenging due to the privacy concerns of the data.' to indicate that it's challenging to seek global flatness in the federated learning setting. This statement applies equally to FedSAM. Thus, FedSAM seeks local flatness to avoid violating this requirement, while GFM tried to seek global flatness by virtue of GCA, which adhere the principle as well.

---

> > ### Author Response · Authors · 2024-11-25
> > **Kind Reminder to Review Our Rebuttal**
> >
> > Dear Reviewer E99p,
> >
> > We sincerely appreciate the time and effort you have dedicated to providing constructive feedback on our submission. As the deadline for discussion and paper revisions approaches, we would like to provide a brief summary of our responses and updates:
> >
> > - Further clarification on the differences between our approach, FedSAM, and Suzuki (2022).
> > - The effects of ablating the augmentation model.
> > - Additional explanation of our motivation, along with a reorganization of the related work and introduction sections.
> > - Updates to Table 1 for improved clarity.
> > - Detailed steps for Figure 2.
> > - Further explanation of the privacy concerns.
> >
> > **Would you mind checking our responses and confirming if you have any additional questions? We welcome any further comments and discussions!**
> >
> > Best Regards,
> >
> > The authors

---

> > > ### Comment · Reviewer_E99p · 2024-11-25
> > > **Thank you for authors' response**
> > >
> > > Thank you for addressing my concerns regarding this paper. However, I still have the following questions:
> > >
> > > 1.As the authors claim, “When the augmentation network is removed, our method reduces to FedAvg+SAM, which is essentially FedSAM as discussed in line 480,” does this mean that the proposed GFM is fundamentally based on FedSAM combined with the augmentation network? If so, my major concern remains that the observed performance improvement may be primarily attributed to the augmentation network.
> > >
> > > 2.Regarding the ablation study results, can FedSAM be considered equivalent to GFM without the augmentation network?
> > >
> > > I think this paper falls on the borderline. Given my major concern, I intend to maintain my score of 5.

---

> > > > ### Author Response · Authors · 2024-11-25
> > > > **Further Responses to Reviewer E99p**
> > > >
> > > > Thank you for your reply and for your efforts in reviewing our work. We would like to further address your concerns.
> > > >
> > > > First, we would like to reiterate the overall motivation of our work and the functionality of each component. Our work aims to achieve **flat minima of the global model** on the **global data distribution**. However, in the FedDG setting, we can only access **local models** trained on **local distributions**. To bridge this gap, as illustrated in Theorem 2, we require a **surrogate for the global data** and aim to achieve **flat minima of the local model** on this **surrogate**. Thus, two components are needed in our method: one component (**GCA**) is used to **generate a surrogate for the global data**, and the other component (**SAM**) is adopted to compute the **flatness of the local model on the surrogate**.
> > > >
> > > > > As the authors claim, “When the augmentation network is removed, our method reduces to FedAvg+SAM, which is essentially FedSAM as discussed in line 480,” does this mean that the proposed GFM is fundamentally based on FedSAM combined with the augmentation network?
> > > > >
> > > > > Regarding the ablation study results, can FedSAM be considered equivalent to GFM without the augmentation network?
> > > >
> > > > Based on the above clarification, the answers to these questions are **yes**. However, as emphasized in our previous response (to W1 & Q2), we consider the compatibility with existing components a contribution of our work. Furthermore, we demonstrate that the **SAM component can be replaced with SWAD, making GFM (SWAD) independent of FedSAM**. Similarly, we believe that the **GCA component can be replaced with other data generators**, such as GANs, while adhering to the **objective in Eq. (14)**. We chose the augmentation network for its lightweight nature.
> > > >
> > > > > If so, my major concern remains that the observed performance improvement may be primarily attributed to the augmentation network.
> > > >
> > > > Regarding your major concern, we believe this may not be entirely accurate, and we would like to clarify it from three perspectives:
> > > >
> > > > | Method |  Digits   |   PACS    | OfficeHome | TerraInc  |   Avg.    |
> > > > | ------ | :-------: | :-------: | :--------: | :-------: | :-------: |
> > > > | FedAvg |   67.46   |   82.56   |   64.82    |   44.23   |   64.77   |
> > > > | FedSAM |   66.67   |   83.36   |   65.28    |   45.16   |   65.12   |
> > > > | GCA    |   69.65   |   83.91   |   64.64    |   44.20   |   64.93   |
> > > > | GFM    | **69.72** | **84.46** | **65.57**  | **46.02** | **66.44** |
> > > >
> > > > - **The results cannot be inferred solely from the table:** On the OfficeHome and TerraInc datasets, using the augmentation network (GCA) alone results in performance that is only comparable to the FedAvg baseline.
> > > > - **It is expected and desirable for the augmentation network to impact performance:** Changes in data distribution inherently influence generalization performance. GCA evaluates the effect of training on surrogate global data, which should definitely  impact the final performance. This aligns with its purpose as a component that creates a surrogate data distribution.
> > > >
> > > > - **The appearance of GCA and general augmentation method is fundamentally different.:** General augmentation methods expand the local data distribution to its its vicinity distribution or its neighborhood. Then we sample from the vicinity distribution by randomness. However, in GCA, the distribution of the augmented samples is not at random at all. Specifically, we use the global and local model to guide the generation of the augmented samples. The **relationship between GCA and general augmentation methods is analogous to the relationship between adversarial samples and random permutation samples**—they serve distinct purposes and are not equivalent. Experimental results further illustrate this distinction. In the table, we compare GCA with two general augmentation methods with color and geometry transformations. While general augmentation methods perform relatively well on the PACS dataset, they perform poorly on datasets like OfficeHome and TerraInc without guidance. This highlights the different appearance of GCA and general augmentation methods.
> > > >
> > > > | Method | Digits-DG |   PACS    | OfficeHome | TerraInc  |   Avg.    |
> > > > | ------ | :-------: | :-------: | :--------: | :-------: | :-------: |
> > > > | RA     |   68.81   | **84.71** |   63.39    |   42.77   |   64.92   |
> > > > | AA     |   69.43   |   84.60   |   63.66    |   43.39   |   65.27   |
> > > > | GCA    | **69.65** |   83.91   | **64.64**  | **44.20** | **65.60** |
> > > >
> > > > We hope this clarification provides a clearer perspective on the role of the augmentation model in our work and the relationship between the components of GFM and other approaches. We would greatly appreciate it if you could reassess our work in light of our new responses. If there is any additional information or clarification we could provide to further improve your evaluation of our paper, we would sincerely appreciate the opportunity. Thank you again.

---

> > > > > ### Author Response · Authors · 2024-12-02
> > > > > **We anticipate your feedback!**
> > > > >
> > > > > Dear Reviewer E99p,
> > > > >
> > > > > Thank you very much for your time and comments.
> > > > >
> > > > > We understand that you may have a busy schedule, but as the discussion deadline is fast approaching, we now have just one day remaining.
> > > > >
> > > > > Would you mind checking our response and confirming whether your concerns are addressed and whether you have any further questions?
> > > > >
> > > > > Thank you again for your attention and consideration.
> > > > >
> > > > > Best regards,
> > > > >
> > > > > The authors.

---

### Official Review · Reviewer_Njhx · 2024-11-10

**Soundness:** 2
**Presentation:** 3
**Contribution:** 2
**Rating:** 3
**Confidence:** 3

**Summary:**

The authors adapt the work that ties flat minima to domain generalization in the context of Federated Learning. They show that the prior work in the centralized case does not naturally extend to federated learning. They then move on to build an intuitive case for its extension and propose a new method that utilizes sharpness-aware minimization and data augmentation to simulate global data distribution during client optimization.

**Strengths:**

The paper is easy to read, and the authors explain the concepts in a linear fashion, making it intuitive to understand the decisions they make while formulating their proposed solution. The authors also clearly identify the shortcomings of existing works and the reasons why they would fail to extend to FL, thus proving the necessity of this work.

**Weaknesses:**

While the paper writing is good, the problem is relevant, and the proposed solution is headed in a promising direction, several weaknesses and gaps in the study need to be addressed. The main concerns are listed below:
1. The ablation study in Table 2 shows that the main benefits are achieved by using SAM and/or GCA. This conflicts the key contribution of this work, i.e., GFM. Particularly, the authors mention "combining them can complement each other’s strengths" in line 516, but looking at the table the same cannot be inferred.
2. Also from table 2, it can be seen that FedAvg+GCA wins in two of the 4 cases bringing up the next question: What is the impact of data augmentation in general when compared to the adversarial data-augmentation that the author's propose.
3. The study in section 5.4 is very thorough and quite informative. The main question that the paper does not address is two-fold
	- the statement "aggregation helps generalization." in line 191 needs either proof or a citation from prior work that proves it. While there is a citation in the paper that advocates that generalization can be achieved via flatness, one cannot infer that flatness is the necessary condition for generalization"
	- secondly the authors need to prove that this statement is further not affected by the relaxation in assumption 1.
	- Overall, while it irrefutable that authors are able to generate improved performances, it is not convincing that the reason for improvements can be attributed totally to flatness of the minima of the global model.
4. Apart from comparing the methods based on performance that authors should also present the comparison in terms of computational and space complexity, e.g., plot comparing FLOPs vs performance or number of parameters shared vs performance, etc.

**Questions:**

1. Authors mention that GFM = FedAvg + SAM + GCA. However, FedSAM is FedAvg + SAM. Could the authors clarify their contribution with respect to FedSAM?
2. Currently the authors are working in the restrictive setting that each client has one domain each. How do they see their method adapting if each client has access to more than one but not all the domains?
3. Is it right to infer that Theorem 1 is a restatement and not an original contribution?
4. Could the authors mention some citations that use a similar assumption as "assuming aggregation helps generalization" on lines 191-192?
5. In Algorithm 1:
	- Can $c>e$? How do the authors choose the values of these hyperparameters? It is quite important to understand this especially since this is an adversarial optimization that is often unstable under the slightest change of hyperparameters.
	- How many models need to be maintained by a client to run this optimization?
	- Could the authors expand line 11 to include loop oversampling, gradient estimation, and update rule? Are the $\phi_i$s also aggregated?
6. Authors mention that the SAM optimizer uses $\gamma=0.02$. How do they arrive at this value? Could they talk about how to interpret this value or how it might impact the training?

---

> ### Author Response · Authors · 2024-11-22
> **Response to Reviewer Njhx [1/5]**
>
> ## W1 & W3.3
>
> > W1:  The ablation study in Table 2 shows that the main benefits are achieved by using SAM and/or GCA. This conflicts the key contribution of this work, i.e., GFM. Particularly, the authors mention "combining them can complement each other’s strengths" in line 516, but looking at the table the same cannot be inferred.
> >
> > W3.3:  Overall, while it irrefutable that authors are able to generate improved performances, it is not convincing that the reason for improvements can be attributed totally to flatness of the minima of the global model.
>
> Thank you for your careful review and constructive feedback. We understand your concerns and acknowledge that the ablation study in Table 2 may not fully illustrate the effectiveness of GFM. In GFM, global flatness is **highly coupled** with the proposed GCA component. Additionally, as an augmentation strategy, **GCA can influence generalization performance**, further complicating the analysis of global flatness. To address this, we have conducted extensive ablation studies on GFM across **a wide range of datasets**, providing deeper insights into global flatness, as detailed in Section 4.5 of the revised manuscript.
>
> | Method |  Digits   |   PACS    | OfficeHome | TerraInc  |   Avg.    |
> | ------ | :-------: | :-------: | :--------: | :-------: | :-------: |
> | FedAvg |   67.46   |   82.56   |   64.82    |   44.23   |   64.77   |
> | FedSAM |   66.67   |   83.36   |   65.28    |   45.16   |   65.12   |
> | GCA    |   69.65   |   83.91   |   64.64    |   44.20   |   64.93   |
> | GFM    | **69.72** | **84.46** | **65.57**  | **46.02** | **66.44** |
>
> Combining the new table with the original one, we have the following observations to answer your question:
>
> - **The effectiveness of global flatness is evident in some cases:** it's worth noting that, in the OfficeHome and TerraInc datasets, GCA alone does not enhance generalization performance, which underscores the importance and effectiveness of the claimed improved flatness of the global model.
> - **GCA or FedSAM alone can achieve the best performance in certain domains of specific datasets:** This typically occurs when either flatness or augmentation negatively impacts generalization performance. In such cases, removing the problematic component yields the best results. While the underlying causes of these failure modes require further investigation, it is encouraging to note that GFM consistently improves the performance in the average meaning.
>
> Regarding the statement "combining them can complement each other’s strengths," we intended to convey that when one component underperforms, the other compensates, resulting in an overall performance improvement compared to the baseline without both of them. However, to avoid potential confusion, we have removed this statement from the revised manuscript.

---

> ### Author Response · Authors · 2024-11-22
> **Response to Reviewer Njhx [2/5]**
>
> ## W2
>
> > Also from table 2, it can be seen that FedAvg+GCA wins in two of the 4 cases bringing up the next question: What is the impact of data augmentation in general when compared to the adversarial data-augmentation that the author's propose.
>
> We would like to clarify that in the following table. We has conducted comparisons with several data augmentation methods: RA (RandAugment [1]), AA (AutoAugment [2]), and Cutout [3]. These methods represent widely-used general data augmentation strategies for images. Notably, RA and AA consist of color and geometric transformations, similar to GCA.
>
> | Method | Digits-DG |   PACS    | OfficeHome | TerraInc  |   Avg.    |
> | ------ | :-------: | :-------: | :--------: | :-------: | :-------: |
> | RA     |   68.81   | **84.71** |   63.39    |   42.77   |   64.92   |
> | AA     |   69.43   |   84.60   |   63.66    |   43.39   |   65.27   |
> | Cutout |   60.75   |   81.90   |   63.42    |   41.43   |   61.88   |
> | GCA    | **69.65** |   83.91   | **64.64**  | **44.20** | **65.60** |
>
> These results clearly demonstrate the effectiveness of GCA in the context of federated domain generalization. For the PACS dataset, we further illustrate that our methods are compatible with these general augmentation strategies, as shown below:
>
> | Method |    Art    |  Cartoon  |   Photo   |  Sketch   |   Avg.    |
> | ------ | :-------: | :-------: | :-------: | :-------: | :-------: |
> | RA     |   83.04   |   78.46   |   93.83   |   83.50   |   84.71   |
> | +GCA   | **83.40** | **78.83** | **94.17** |   86.28   | **85.67** |
> | AA     |   82.57   |   77.43   |   93.91   |   84.47   |   84.60   |
> | +GCA   |   81.64   |   78.77   |   93.43   | **87.44** |   85.32   |
> | Cutout |   76.98   |   77.77   |   92.12   |   80.74   |   81.90   |
> | +GCA   |   79.91   |   77.80   |   91.62   |   84.48   |   83.45   |
>
> ### Reference
>
> [1] Cubuk, Ekin D., et al. "Randaugment: Practical automated data augmentation with a reduced search space." *Proceedings of the IEEE/CVF conference on computer vision and pattern recognition workshops*. 2020.
>
> [2] Cubuk, Ekin D., et al. "Autoaugment: Learning augmentation strategies from data." *Proceedings of the IEEE/CVF conference on computer vision and pattern recognition*. 2019.
>
> [3] DeVries, Terrance. "Improved Regularization of Convolutional Neural Networks with Cutout." *arXiv preprint arXiv:1708.04552* (2017).
>
> ## W3.1 & W3.2 & Q4
>
> > W3.1: The statement "aggregation helps generalization." in line 191 needs either proof or a citation from prior work that proves it. While there is a citation in the paper that advocates that generalization can be achieved via flatness, one cannot infer that flatness is the necessary condition for generalization".
> >
> > Q4: Could the authors mention some citations that use a similar assumption as "assuming aggregation helps generalization" on lines 191-192?
>
> The statement "aggregation helps generalization" refers to Assumption 1 in our paper (line 187).  If I understand your concern correctly, you are asking **whether flatness contributes to generalization**. There is an insightful study [4] in the literature that discusses the relationship between flatness, generalization, and SAM across various architectures and data distributions. We briefly introduce its conclusions below:
>
>
> - **Flatness is neither a necessary nor sufficient condition for generalization:** For instance, sharp minima can sometimes generalize well, while flat minima may generalize poorly.
>
> - Situations fall into three regimes: (1) Flattest minimizers of training loss provably generalize and sharpness minimization algorithms find generalizable models. (2) There exists a flattest minimizer that generalizes poorly. Also, sharpness minimization algorithms fail to find generalizable models. (3) There exist flattest minimizers that do not generalize but the sharpness minimization algorithm still finds the generalizable flattest model empirically.
>
> While theoretical results cannot guarantee that "flatness helps generalization" under all architectures and data distributions, empirical evidence demonstrates its effectiveness across a variety of tasks and model architectures. Therefore, pursuing flatness remains a reasonable strategy to achieve better generalization in the context of the FedDG task.
>
> > W3.2: Secondly the authors need to prove that this statement is further not affected by the relaxation in assumption 1.
>
> Assumption 1 pertains to the **relationship between models before and after aggregation** in the FedDG setting and is not directly related to the concept of flatness. Therefore, we believe it is not relevant to the statement "flatness helps generalization".
>
> ### Reference
>
> [4] Wen, Kaiyue, Zhiyuan Li, and Tengyu Ma. "Sharpness minimization algorithms do not only minimize sharpness to achieve better generalization." *Advances in Neural Information Processing Systems* 36 (2024).

---

> ### Author Response · Authors · 2024-11-22
> **Response to Reviewer Njhx [3/5]**
>
> ## W4
>
> > Apart from comparing the methods based on performance that authors should also present the comparison in terms of computational and space complexity, e.g., plot comparing FLOPs vs performance or number of parameters shared vs performance, etc.
>
> Thank you for the valuable advice. In the original paper, we compared the running time and space cost of GFM and FedAvg. Following your suggestion, we further refined the table and included three metrics—**running time, space cost of local updates (former element in the space column), and the parameters required for sharing (latter element in the space column)** —to evaluate each method in Table 7 of the revised manuscript.
>
> |          |  Digits-DG  |            |    PACS     |           | OfficeHome  |           |  TerraInc   |           |
> | -------- | :---------: | :--------: | :---------: | :-------: | :---------: | :-------: | :---------: | :-------: |
> |          |    time     |   space    |    time     |   space   |    time     |   space   |    time     |   space   |
> | FedAvg   |     $V$     |   $M,M$    |     $V$     |   $M,M$   |     $V$     |   $M,M$   |     $V$     |   $M,M$   |
> | Scaffold | $\approx V$ |  $4M,2M$   | $\approx V$ |  $4M,2M$  | $\approx V$ |  $4M,2M$  | $\approx V$ |  $4M,2M$  |
> | FedDyn   | $\approx V$ |   $3M,M$   |   1.34$V$   |  $3M,M$   | $\approx V$ |  $3M,M$   | $\approx V$ |  $3M,M$   |
> | MOON     | $\approx V$ |   $3M,M$   |   1.38$V$   |  $3M,M$   | $\approx V$ |  $3M,M$   | $\approx V$ |  $3M,M$   |
> | FedSAM   |   1.08$V$   |   $M,M$    |   1.73$V$   |   $M,M$   |   1.43$V$   |   $M,M$   |   1.35$V$   |   $M,M$   |
> | FedGAMMA |   1.09$V$   |  $4M,2M$   |   1.70$V$   |  $4M,2M$  |   1.51$V$   |  $4M,2M$  |   1.32$V$   |  $4M,2M$  |
> | FedSMOO  |   1.22$V$   |  $5M,2M$   |   2.00$V$   |  $5M,2M$  |   1.72$V$   |  $5M,2M$  |   1.48$V$   |  $5M,2M$  |
> | FedSR    | $\approx V$ |   $M,M$    | $\approx V$ |   $M,M$   | $\approx V$ |   $M,M$   | $\approx V$ |   $M,M$   |
> | GA       | $\approx V$ |   $M,M$    | $\approx V$ |   $M,M$   | $\approx V$ |   $M,M$   | $\approx V$ |   $M,M$   |
> | GFM      |   1.42$V$   | $25.91M,M$ |   2.86$V$   | $2.25M,M$ |   2.01$V$   | $2.25M,M$ |   1.55$V$   | $2.12M,M$ |
> | GFM (GA) |   1.45$V$   | $25.91M,M$ |   2.92$V$   | $2.25M,M$ |   1.95$V$   | $2.25M,M$ |   1.62$V$   | $2.12M,M$ |
>
> From these results, we can conclude the following:
>
> - The additional parameters introduced by the augmentation model are relatively small.
> - The actual running time increases to approximately 1.5 to 1.7 times that of the FedSAM baseline.
> - Among methods  (FedGAMMA, FedSMOO, and GFM) designed to achieve flatter minima for the global model, our method imposes the **lowest space constraints**.

---

> ### Author Response · Authors · 2024-11-22
> **Response to Reviewer Njhx [4/5]**
>
> ## Q1
>
> > Authors mention that GFM = FedAvg + SAM + GCA. However, FedSAM is FedAvg + SAM. Could the authors clarify their contribution with respect to FedSAM?
>
> The focus of the paper is to **seek flatter minima of the global model**. As demonstrated in Theorem 2, we say the performance on the unseen domain is directly related to the **local flatness of seen clients on augmented samples**. This insight suggests that there is no need to design a dedicated global flatness strategy; instead, we can **combine any existing local flatness optimizer with the proposed GCA**, which we consider a key strength and contribution of our work. Therefore, to make the overall algorithm as simple as possible, we adopt the well-known SAM optimizer in local update.
>
> To further demonstrate, we show in the following that our method could combine with another local flatness optimizer. Specifically, we use SWAD [5] to replace SAM in local training ($n_{converge}=5,n_{tolerance}=6$ and the ratio of tolerance as $0.2$). It is worth noting that, due to time constraints, we have not extensively tuned the hyperparameters, and the results may be suboptimal.
>
> | Method    |    Art    |  Cartoon  |   Photo   |  Sketch   |   Avg.    |
> | --------- | :-------: | :-------: | :-------: | :-------: | :-------: |
> | FedAvg    |   77.41   |   77.82   |   92.67   |   82.35   |   82.56   |
> | FedSAM    |   78.34   | **78.85** |   92.45   |   83.81   |   83.36   |
> | GFM(SWAD) | **80.71** |   78.29   | **93.20** |   84.57   |   84.19   |
> | GFM(SAM)  |   80.34   |   78.27   |   92.53   | **86.70** | **84.46** |
>
> To summary, we clarify the contributions as follows:
>
> - Though the SAM component is shared between GFM and FedSAM, **the focus differs significantly**. GFM emphasizes achieving flatness for the global model, whereas FedSAM is concerned only with local flatness. (It is also worth noting that SAM was not originally proposed in FedSAM.)
> - Our work can be **combined with other local flatness optimizer** to take place of SAM, like SWAD.
> - Finally, we consider the compatibility of reusing established components in addressing new challenges, with accompanying theoretical guarantees, as an important contribution of our work.
>
> ### Reference
>
> [5] Cha, Junbum, et al. "Swad: Domain generalization by seeking flat minima." *Advances in Neural Information Processing Systems* 34 (2021): 22405-22418.
>
> ## Q2
>
> > Currently the authors are working in the restrictive setting that each client has one domain each. How do they see their method adapting if each client has access to more than one but not all the domains?
>
> Theoretically, the applicability of our method primarily depends on whether Assumption 1 is satisfied. In the mentioned case, Assumption 1 remains valid because, in principle, a local model trained on a subset of domains cannot outperform the global model when evaluated across all domains. To further validate it, we conducted experiments on PACS dataset. In this setting, when testing on the photo domain, $A,C,S$  datasets are randomly split into $A_1,A_2,C_1,C_2,S_1,S_2$, which were then recombined as $(A_1, C_1)$, $(A_2, S_1)$, and $(C_2, S_2)$ to represent different clients.
>
> | Method |    Art    |  Cartoon  |   Photo   |  Sketch   |   Avg.    |
> | ------ | :-------: | :-------: | :-------: | :-------: | :-------: |
> | FedAvg |   82.50   |   78.16   | **95.51** |   80.10   |   84.07   |
> | FedSAM |   82.57   | **78.78** |   94.99   |   81.83   |   84.54   |
> | GFM    | **83.29** |   78.27   |   94.99   | **84.71** | **85.32** |
>
> The results above indicate that the mentioned scenario is analogous to the single-domain case.

---

> ### Author Response · Authors · 2024-11-22
> **Response to Reviewer Njhx [5/5]**
>
> ## Q3
>
> > Is it right to infer that Theorem 1 is a restatement and not an original contribution?
>
> Yes, it is right. We stated it in the Line 136 of the paper.
>
> ## Q5
>
> > Q5.1: Can c>e? How do the authors choose the values of these hyperparameters? It is quite important to understand this especially since this is an adversarial optimization that is often unstable under the slightest change of hyperparameters.
>
> $c$ refers to the update interval, and $e$ refers to the iterations of local updates. $c>e$ means during the whole local training, the augmentation model won't be updated, which we don't suggest. To understand the selection of $c$, we conduct parameter analysis in Section 4.6 of the revised version. We briefly introduce its conclusions as follows:
>
> - **The performance improves with more frequent updates of the augmentation model:** the strategy of alternating one iteration of augmentation with one iteration of classification achieves the best performance.
> - **To balance performance and efficiency**, we set $c = 10$ for related experiments.
>
> > Q5.2: How many models need to be maintained by a client to run this optimization?
>
> Each client needs to maintain the received global model $\theta^{r-1}$, the local model $\theta^r_i$, and the augmentation model $\phi_i$.
>
> > Q5.3: Could the authors expand line 11 to include loop oversampling, gradient estimation, and update rule? Are the ϕis also aggregated?
>
> Sure, and we apologize for the error in line 11.In practice, we update $\phi_i$ as shown on the right-hand side of Eq. (14), rather than Eq. (9). Additionally, line 11 does not include loop oversampling; instead, $\phi_i$ is updated solely based on the current batch $X_i$ by augmenting it and feeding it through the prediction model to compute gradients. In the revised version, line 11 has been expanded to lines 11–12 to provide more detail.
>
> ​	*11: Compute $g_{\phi_i}=\nabla_{\phi_i} \hat{\mathcal{E}}_{a(D_i;\phi_i)}(\theta_i) - \hat{\mathcal{E}}_{a(D_i;\phi_i)}(\theta)$ on $X_i$*
>
> ​	*12: Update $\phi_i=\phi_i+\rho_\phi g_{\phi_i}$  		Update augmentation model $\phi_i$*
>
>  $\{\phi_i\}$ do not need to be aggregated, although doing so is feasible. Aggregating them would only affect the initialization of $\phi_i$ in each round.
>
> ## Q6
>
> > Authors mention that the SAM optimizer uses $\gamma=0.02$. How do they arrive at this value? Could they talk about how to interpret this value or how it might impact the training?
>
> The radius $\gamma$ is a critical hyperparameter in SAM-based methods, as it determines the range of model perturbation. The optimal value of $\gamma$ varies across tasks, datasets, and models. So we perform a grid search to determine the appropriate value of $\gamma$ as shown in Section 4.6.

---

> > ### Comment · Reviewer_Njhx · 2024-11-22
> >
> > I thank the authors for taking the time to put together a detailed response.
> >
> > As per my assessment of the work, the authors propose to "seek global flat minima for federated domain generalization". However, the experimental observations diminish this proposal in the context of domain generalization. Particularly, factors (also admitted by the authors above) such as (i) the effectiveness of global flatness is evident in only some cases, (ii) GCA can influence generalization performance, and (iii) Flatness is neither a necessary nor sufficient condition for generalization, make it very difficult to understand clearly: "What is the impact of seeking global minima in context of improving generalization performance?" Moreover, even if we consider empirically that flatness helps improve generalization, it is hard to decouple the impact of GFM from GCA (or SWAD that authors mention above) across datasets convincingly.
> >
> > Given these reservations, I am inclined to keep my current score.

---

> > > ### Author Response · Authors · 2024-11-25
> > >
> > > Thank you for your prompt response and for the effort you’ve put into reviewing our work. We respect your opinion and, while understanding your perspective, we would like to further clarify it.
> > >
> > > The remaining question pertains to decoupling and independently verifying the effects of global flatness and GCA. Changes in data distribution inherently impact generalization performance, making complete disentanglement a persistent challenge. What we can do—and have attempted to do—is to focus on scenarios where GCA alone performs poorly and analyze whether the introduction of global flatness improves performance. These observations provide meaningful evidence supporting the effectiveness of global flatness as a significant contribution.
> > >
> > > While our work didn't achieve perfect disentanglement, we believe this does not diminish the overall value of the proposed method. Furthermore, GCA itself is a key contribution of our work, as it serves to evaluate the impact of training on surrogate global data. This, in turn, establishes a foundation for assessing how incorporating flatness into training on such data influences performance. Importantly, our experimental results consistently demonstrate that GFM enhances performance across datasets, underscoring its practical utility.
> > >
> > > We hope this clarification helps to offer a clearer perspective on the contributions of our work. If there is any additional information or clarification we could provide to further improve your evaluation of our paper, we would sincerely appreciate the opportunity. Thank you again.

---

### Author Response · Authors · 2024-11-22
**General Response by Authors**

# Summary

We thank reviewers for their valuable feedback , and appreciate the great efforts made by all reviewers, ACs, SACs and PCs.

We appreciate that the reviewers have multiple positive impressions of our work, including: (1) focused problem is **of necessity and novelty** (Njhx, iTLS); (2) the proposed solution **is headed in a promising direction** (Njhx, iTLS); (3) **thorough theoretical support** (E99p, hj24, iTLS); (4) **sufficient experiments** (E99p, hj24, iTLS);

We provide a summary of our updates, and for detailed responses, please refer to the feedback of each comment/question point-by-point.

- **Relationship with Related Work:** We clarified the relationship between our work and related studies (e.g., FedSAM) and added an introduction to closely related works in the **Introduction** section. Content from the Related Works section was reorganized, and additional citations were included.
- **Improved Analysis:** We enriched the theoretical analysis of the min-max optimization process and provided detailed explanations in Appendix F. The ablation study is modified for better illustration in Section 4.5. A sensitivity analysis of the SAM radius parameter ($\gamma$) and update interval $c$ was added in Section 4.6. Results on surrogate data effectiveness and mean forgetting rates were added in Appendix C.2.
- **Computational Cost:** A computational efficiency study was included, showing reduced augmentation costs through parallelization and subsampling strategies (see Appendix C.5).
- **Revised Presentation:** Key expressions and equations were refined for clarity.

The above updates in the revised draft (including the regular pages and the Appendix) are highlighted in blue color.

We once again express our gratitude to all reviewers for their time and efforts devoted to evaluating our work. We eagerly anticipate your further responses and are hopeful for a favorable consideration of our revised manuscript.

---

### Meta-Review · Area_Chair_bd2d · 2024-12-17

**Metareview:**

This paper designs GFM that seeks flat global minima in federated domain generalization. The reviewers agreed that it is well written but there is a major lingering concern regarding the overall implications of the work after the discussion phase. I agree with these concerns and hence, assess that the paper does not cross the bar for publication at ICLR. In particular, GFM can be thought of as FedSAM plus an augmentation network called global model-constrained adversarial data augmentation (GCA). Reviewers Njhx and E99p feel that it remains unclear whether the empirical improvements are entirely attributed to GCA. In other words, it remains unconvincing that the reason for improvements can be attributed totally to flatness of the minima of the global model, but perhaps it is due to GCA.

**Additional Comments On Reviewer Discussion:**

There was a robust and healthy exchange between the authors and reviewers in the discussion phase. The authors presented several more experimental results and also tried to explain the origins of the performance improvements, i.e., whether its due to the flatness of the minima of the global model or it is due to GCA. After the exchange, the reviewers remained unconvinced by the authors explanations and maintained that the improvements in performance could simply be due to GCA.

---

### Decision · Program_Chairs · 2025-01-22

Reject